# A Minimalist Example of Edge-of-Stability and Progressive Sharpening

**Liming Liu**
Georgia Institute of Technology
lliu606@gatech.edu

**Zixuan Zhang**
Georgia Institute of Technology
zzhang3105@gatech.edu

**Simon Shaolei Du**
University of Washington
ssdu@cs.washington.edu

**Tuo Zhao**
Georgia Institute of Technology
tourzhao@gatech.edu

## Abstract

Recent advances in deep learning optimization have unveiled two intriguing phenomena under large learning rates: Edge of Stability (EoS) and Progressive Sharpening (PS), challenging classical Gradient Descent (GD) analyses. Current research approaches, using either generalist frameworks or minimalist examples, face significant limitations in explaining these phenomena. This paper advances the minimalist approach by introducing a two-layer network with a two-dimensional input, where one dimension is relevant to the response and the other is irrelevant. Through this model, we rigorously prove the existence of progressive sharpening and self-stabilization under large learning rates, and establish non-asymptotic analysis of the training dynamics and sharpness along the entire GD trajectory. Besides, we connect our minimalist example to existing works by reconciling the existence of a well-behaved "stable set" between minimalist and generalist analyses, and extending the analysis of Gradient Flow Solution sharpness to our two-dimensional input scenario. These findings provide new insights into the EoS phenomenon from both parameter and input data distribution perspectives, potentially informing more effective optimization strategies in deep learning practice.

## 1 Introduction

Deep learning has revolutionized many fields, from computer vision to natural language processing. However, this progress has also posed significant challenges to classical optimization theory. Most classical gradient descent (GD) analysis assumes small learning rates for easing convergence analysis. Consider minimizing a smooth loss function $L(\theta)$ with respect to the parameter $\theta$, classical analyses show that when choosing a learning rate $\eta$ such that $S(\theta) \leq 2/\eta$, where $S(\theta)$ denotes the largest eigenvalue of the Hessian matrix $\nabla^2 L(\theta)$, the optimization is "stable" and the loss function decreases monotonically to guarantee convergence Nesterov [2013].

Recent works such as Cohen et al. [2021], however, have observed that such a stability assumption does not hold when training modern neural networks with GD. In particular, they summarize two specific phenomena: The first one is called *"Progressive Sharpening"*(PS), that is, $S(\theta)$, which is also referred to as the "sharpness" in Cohen et al. [2021], keeps increasing until it reaches the instability threshold $2/\eta$ during training; The second one is called *"Edge of Stability"*(EoS), that is, the sharpness hovers at the instability threshold $2/\eta$ after progressive sharpening, with the loss decreases continuously and nonmonotonically. These two phenomena undoubtedly challenge the classical analyses and have already attracted the attention of many researchers.

39th Conference on Neural Information Processing Systems (NeurIPS 2025).

Current research on understanding these phenomena has developed along two lines. The first line purses a so-called "generalist analysis" frameworks that, while being generic, rely on hard-to verify assumptions. For example, Li et al. [2022b] analyze GD for training two-layer wide neural networks. By characterizing the norms of the second-layer weights, they prove a four-stage behavior, covering the PS and EoS phenomena. However, their analysis requires assumptions that are hard to verify for two-layer neural networks, e.g., the sharpness being upper bounded, and only works for extremely wide settings, which diverges from practical scenarios. Damian et al. [2022] prove a similar four-stage behavior based on a general loss function, and their analysis also relies on hard-to-verify assumptions, such as the existence of progressive sharpening and the existence of a certain well-behaved "stable set", which is doubted and showed badly-behaved for scalar networks in [Kreisler et al., 2023]. Overall speaking, such analysis provide results similar to real experiments, but their hard-to-verify assumptions significantly restrict their applicability to practice. Relaxing these assumptions is also extremely challenging.

The second line focuses on minimalist examples, offering more concrete and intuitive insights without requiring specific theoretical assumptions. For instance, Zhu et al. [2022] study $2d$ slices of a 4-layer linear scalar network and prove convergence to a minimum with sharpness slightly below to $2/\eta$, and Wang et al. [2023] consider the behavior of sharpness with GD on a special class of 2-layer scalar networks (with nonlinear activation) in the form of $F(xy)$. However, these works suffer from two drawbacks: (1) They can only characterize the asymptotic sharpness of the converged minimum. Due to the lack of analyzing sharpness for the entire trajectory, they cannot provide more desirable nonasymptotic guarantees for the PS and EoS stages; (2) The setting of scalar networks is over-simplified compared with practice, and the obtained results cannot explain the role of the input data dimension in the EoS stage.

Besides, another notable example of the minimalist analysis is Kreisler et al. [2023], which analyze a variant of sharpness. Specifically, they introduce a concept of Gradient Flow Solution (GFS), and prove a monotonic decrease in the sharpness of GFS at the EoS stage for scalar networks. In addition to the aforementioned drawbacks, Kreisler et al. [2023] also suffer from another drawback: The sharpness of GFS does not directly transfer to that of GD trajectory, thus providing no immediate explanation for the EoS phenomenon.

Here we aim to address the limitations of the minimalist analysis by providing a more nontrivial example: A two-layer neural network of width one with a two-dimensional input – in particular, one input dimension is relevant to the response, and the other one is irrelevant. We establish nonasymptotic analysis of the training dynamics along the entire trajectory: (1) We prove the existence of progressive sharpening and self-stabilization under large learning rates; (2) We provide sharpness guarantees for the entire trajectory, showing that GD trajectory will never exceed a sharpness upper bound; (3) We prove that the non-monotonically decreasing loss is essentially monotonically decreasing when projected to the only relevant dimension. Through such theory, we provides new insights of why EoS happens from the perspective of both parameters and input data distribution.

Moreover, we highlight two connections of our theory to existing works: (1) We reduce the gap on the existence of a well-behaved "stable set" between the minimalist and generalist analyses. Specifically, Kreisler et al. [2023] prove that the stable set hypothesized in Damian et al. [2022] can be disjoint in the scalar networks studied by Zhu et al. [2022], Kreisler et al. [2023], which essentially violates the assumption of Damian et al. [2022]. Therefore, the projected GD considered in Damian et al. [2022] cannot smoothly decrease the loss toward zero. In contrast, we prove that in our considered two-layer neural network with the two-dimensional input admits a nontrivial well-behaved set, which is the subset of the stable set defined in Damian et al. [2022]. This indicates a potential separation between the bivariate and scalar inputs for linear networks; (2) We extend the analysis of Kreisler et al. [2023] and provide the monotonic decrease of GFS sharpness for our considered two-layer neural network with the two-dimensional input.

## 2 Related Works

The asymptotic property of GD sharpness was first mentioned in Wu et al. [2018] as an empirical observation, that is, the minimal sharpness that the GD trajectory with learning rate $\eta$ ultimately converges to is always around $\frac{2}{\eta}$. Cohen et al. [2021] made a comprehensive empirical study on the sharpness of the entire GD trajectory. To be more specific, they summarized two phenomena:

"progressive sharpening" and "edge of stability", which means the sharpness of gradient descent with a learning rate $\eta$ will first increase to $\frac{2}{\eta}$ and then stabilize at such a scale during the entire training process. They also illustrated that the training loss of GD with a learning rate $\eta$ can non-monotonically decrease, even when the stable condition, sharpness $\lambda \leq \frac{2}{\eta}$ (where $\eta$ is the learning rate), is not satisfied. The non-monotonic decay property of the training loss with GD has also been observed in various other settings [Wu et al., 2018, Arora et al., 2018, Xing et al., 2018, Jastrzebski et al., 2020, Lewkowycz et al., 2020, Wang et al., 2021, Li et al., 2022a].

Recently, several works have attempted to comprehend the mechanism behind EoS with different loss functions under various assumptions [Ahn et al., 2022, Ma et al., 2022, Arora et al., 2022, Lyu et al., 2022, Li et al., 2022b, Zhu et al., 2022]. From a landscape perspective, Ma et al. [2022] defined a special subquadratic property of the loss function and proved that EoS occurs based on this assumption. Ahn et al. [2022] followed this landscape property and studied the unstable convergence behavior of GD. Both Arora et al. [2022] and Lyu et al. [2022] investigated the implicit bias on the sharpness of GD in some general loss function.

There are also some works trying to investigate EoS phenomenon on highly simplified settings. Zhu et al. [2022] proved the asymptotic sharpness of the converged minimum will be close to $\frac{2}{\eta}$. Wang et al. [2023] consider the behavior of sharpness with GD on a special class of 2-layer scalar networks (with nonlinear activation) in the form of $F(xy)$, and also get a asymptotic result and show some other behavior beyond EoS in their setting. Agarwala et al. [2022] investigate second order regression models, get a asymptotic result similar to Zhu et al. [2022], and a result loosely related to "progressive sharpening". Chen and Bruna [2023] investigate the behavior beyond edge of stability in various simplified examples. Kreisler et al. [2023] consider the setting of scalar networks, and show a new concept gradient flow solution (GFS) will decrease during EoS.

Another line of work [Lewkowycz et al., 2020, Wang et al., 2021] focuses on the implicit bias introduced by large learning rates. Lewkowycz et al. [2020] first proposed the "catapult phase" , a regime similar to the EoS, where loss does not diverge even if the sharpness is larger than $\frac{2}{\eta}$. Wang et al. [2021] analyze the balance effect of GD with a large learning rate for matrix factorization problems. More recently, Li et al. [2022b] provided a theoretical analysis of the sharpness along the gradient descent trajectory in a highly overparameterized two-layer linear network setting under some hard-to-verify assumptions during the training process. Damian et al. [2022] followed Li et al. [2022b] to develop a general theory of self-stabilization also under some hard-to-verify assumptions. Moreover, they proposed a concept called "constrained trajectory" to show that the trajectory of GD with large learning rate deviates from the gradient flow, which was firstly observed by Jastrzebski et al. [2020] and confirmed by Cohen et al. [2021].

# 3 Setup

In this paper, we study a regression problem with two-dimensional input $x = (x_1, x_2)^\top \in \mathbb{R}^2$ and scalar response $y = f^*(x) \in \mathbb{R}$. For analytical simplicity, we suppose

$$\begin{pmatrix} x_1 \\ x_2 \end{pmatrix} \sim N\left(0, \begin{pmatrix} \lambda_1 & 0 \\ 0 & \lambda_2 \end{pmatrix}\right), \text{ and } f^*(x) = x_2,$$

We suppose $\frac{\lambda_1}{\lambda_2} \geq K$, where $K$ is a large enough absolute constant. Here $x_1$ is an irrelevant feature (or noise) with large scale, while small-scale $x_2$ fully determines the response $y$.

Although the setting appears relatively simple, using the rotation invariance property of GD on neural networks, which means that rotating the input data by an orthogonal transformation leaves the gradient descent dynamics unchanged, we can extend this setting to higher-dimensional spaces with non-diagonal data covariance matrix. In addition, exploiting parameter symmetry allows us to extend this framework to handle more than two features. See Appendix C for details.

To learn the target function $f^*$, we use a two-layer width-one linear network with weights $\theta = (\alpha, \beta_1, \beta_2)^T \in \mathbb{R}^3$:

$$f(x; \theta) = \alpha\beta_1 x_1 + \alpha\beta_2 x_2. \tag{3.1}$$

Then the population square loss is given by

$$L(\theta) = \frac{1}{2}\mathbb{E}_{x,y}\left[(y - f(x; \theta))^2\right] = \frac{1}{2}\lambda_1(\alpha\beta_1)^2 + \frac{1}{2}\lambda_2(\alpha\beta_2 - 1)^2. \tag{3.2}$$

The Hessian matrix $H(\theta)$ of $L(\theta)$ can be written as:

$$H(\theta) = \begin{pmatrix} \lambda_1\beta_1^2 + \lambda_2\beta_2^2 & 2\lambda_1\alpha\beta_1 & 2\lambda_2\alpha\beta_2 - \lambda_2 \\ 2\lambda_1\alpha\beta_1 & \lambda_1\alpha^2 & 0 \\ 2\lambda_2\alpha\beta_2 - \lambda_2 & 0 & \lambda_2\alpha^2 \end{pmatrix}.$$

**Definition 3.1** (Sharpness of $L(\theta)$). We define the largest eigenvalue of $H(\theta)$ as the sharpness parameter of $L(\theta)$. We denote it by $S(\theta)$.

Our setting is partly motivated by Rosenfeld and Risteski [2023], which demonstrated that the oscillations observed during the EoS stage in image classification tasks are primarily driven by large-magnitude features in the input data. In particular, such features, such as the background color in CIFAR-10 images, exhibit little correlation with the true labels and effectively act as noise during training. Building on their observations, our work provides rigorous theoretical foundations that formalize and clarify their informal explanations, demonstrating how the EoS phenomenon and the behaviors hypothesized in Rosenfeld and Risteski [2023] emerge. A more detailed discussion is provided in Appendix H.

We consider the following constrained optimization problem in our setting:

$$\min_{\theta \in \Theta} L(\theta) \quad \text{where} \quad \Theta = \{\theta : |\beta_1| \leq c\}$$

The threshold $c = \frac{1}{2}(\lambda_2/\lambda_1)^{1/4}$ is chosen based on the condition number $\lambda_2/\lambda_1$, which reflects the scale disparity between input features. In our setting, $\beta_1$ is associated exclusively with the high-variance, noise-dominated feature $x_1$, while $\beta_2$ contributes to fitting the informative signal. As shown in (3.1), increasing $|\beta_1|$ disproportionately amplifies the effect of noise without improving alignment with the target. In contrast, $\alpha$ scales both $\beta_1$ and $\beta_2$ simultaneously, and thus amplifies noise and signal together, preserving the relative structure of the model's response.

In our setting, the instability in training is caused by large-scale noise. However, when $|\beta_1|$ is too large, the noise is excessively amplified, and the properties of the dynamics go beyond the limits of existing theoretical tools, as current optimization theories mainly focus on convergence behavior and lack appropriate tools for handling instability. Therefore, although we strive to use new methods to analyze the unstable optimization process, we must impose constraints to ensure it does not exceed the theoretical limits. Our numerical experiments in Figure 1 show that even on this constrained problem, the optimization process remains highly unstable. In Section G, we demonstrate that the behavior of the constrained optimization process is consistent with that of the unconstrained one.

The update rule for GD with learning rate $\eta$ in this constrained problem is:

$$\theta(t+1) = \Pi_\Theta \left( \theta(t) - \eta \nabla L(\theta(t)) \right),$$

where $\Pi$ is the projection function. We initialize the weights within the initialization set $\mathcal{X}(\eta)$, which is defined as the set of $\theta$ satisfying

$$\sqrt{\frac{1.1}{\lambda_1\eta}} \leq \alpha \leq \sqrt{\frac{2}{\lambda_1\eta}}, \quad \max\{\frac{\sqrt{6\eta\lambda_1}}{20}, \frac{3}{20\alpha}, \alpha\} \leq \beta_2 < \frac{1}{\alpha},$$

$$\frac{\lambda_2\beta_2}{500\lambda_1\alpha}(1 - \alpha\beta_2) \leq \beta_1^2 \leq \frac{\lambda_2\beta_2}{\lambda_1\alpha}(1 - \alpha\beta_2).$$

Our analyses focus on sufficiently large learning rate $\eta \in [\frac{2}{\lambda_1}, \frac{1}{10\sqrt{\lambda_1\lambda_2}}]$, where $\mathcal{X}(\eta)$ is nonempty. Notably, this initialization allows us to explore the training dynamics across a wide range of learning rates, all starting from the same initial point. Additional details are provided in Appendix D.

The GD dynamics of our model exhibits interesting EoS phenomena. As shown in Figure 1, we observe that while the loss decreases over long timescales, it exhibits non-monotonic behavior with periodic spikes. Meanwhile, the sharpness grows and oscillates around $2/\eta$, with rapid alternation between progressive sharpening and self-stabilization phases. These characteristics align with the EoS phenomena observed in Cohen et al. [2021] and Damian et al. [2022], and extend beyond the scalar network setting studied in Zhu et al. [2022]. We provide further discussion in Section 6.1.

## 4 Theoretical Results

In this section, we develop a nonasymptotic analysis of the GD dynamics throughout the entire trajectory. In Section 4.1, we identify three distinct phases in the GD dynamics, proving the existence

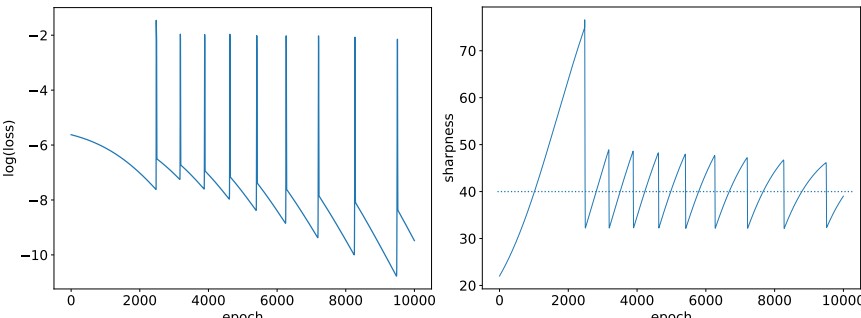

Figure 1: Set $\lambda_1 = 100$ and $\lambda_2 = 0.01$. We train our model with $\eta = 1/20$ for 10000 iterations.

of progressive sharpening and self-stabilization. Furthermore, in Section 4.2, we establish that the loss (non-monotonically) converges to zero, with a rate depending explicitly on the input data variance.

In our analysis, we use learning late $\eta \in [\frac{2}{\lambda_1}, \frac{1}{10\sqrt{\lambda_1 \lambda_2}}]$, and initialization within the set $\mathcal{X}(\eta)$ unless otherwise specified. Our choice of $\eta$ is sufficiently large to guarantee the occurrence of EoS.

## 4.1 Gradient Descent Dynamics

To begin with, we present the three distinct phases identified in the GD dynamics.

**Phase 1: Progressive sharpening before EoS.** The loss $L$ decreases monotonically, while the loss sharpness gradually increases, yet remains below the stability threshold $2/\eta$.

**Phase 2.1: Progressive sharpening during EoS.** As the dynamics enters the EoS (Edge of Stability) stage, the sharpness continues to increase monotonically and exceeds the stability threshold $2/\eta$.

**Phase 2.2: Self-stabilization during EoS.** In this phase of the EoS stage, the dynamics self-stabilize as the sharpness decreases monotonically until it falls below $2/\eta$.

In addition to proving the existence of these phases, we also show that throughout the entire training process, the sharpness is controlled within a range close to $2/\eta$, and that gradient descent eventually converges to a minimum whose sharpness is strictly upper bounded by $2/\eta$.

**Theorem 4.1** (Global Convergence). For any $\delta > 0$ and $\epsilon > 0$, there exists a time $T(\delta, \epsilon)$, such that for any $t \geq T(\delta, \epsilon)$, we have

$$L(\theta(t)) \leq \epsilon \quad \text{and} \quad S(\theta(t)) \leq \frac{2 + \delta}{\eta}. \tag{4.1}$$

To rigorously characterize the GD dynamics, we first present the following properties of the parameters $\theta(t) = (\alpha(t), \beta_1(t), \beta_2(t))$:

**Lemma 4.2.** For all $t \geq 0$, use $v(t)$ to denote the eigenvector corresponding to the largest eigenvalue of $H(\theta(t))$, we have:

**(i)** $\lambda_1 \alpha^2(t) \leq S(\theta(t)) \leq 1.12\lambda_1 \alpha^2(t)$;  **(ii)** $|\cos(v(t), (0, 1, 0))| > 0.9$;  **(iii)** $\beta_2(t + 1) > \beta_2(t)$.

We make the following remarks for Lemma 4.2:

**Sharpness dominated by** $\alpha$  Property (i) demonstrates that $\lambda_1 \alpha^2$ dominates the sharpness $S(\theta)$ along the trajectory, suggesting that analyzing $\alpha$ suffices to reveal the dynamics of sharpness.

**Instability of** $\beta_1$  The top eigenvector $v(t)$ of $H(\theta(t))$ identifies the direction of maximum curvature in $L(\theta(t))$. When the sharpness is as large as $2/\eta$, updates along $v(t)$ can induce oscillatory instability in the GD dynamics. Property (ii) reveals that the oscillating direction $v(t)$ closely aligns with $\beta_1$. Furthermore, we will show that $\beta_1$ induces spikes in the loss $L$ in Section 4.2.

**Monotonic Increase of** $\beta_2$  In contrast to $\beta_1$, $\beta_2$ is a stable direction. This stability difference arises from their respective second-order partial derivatives: $\frac{\partial^2 L}{\partial \beta_2}(\theta) = \lambda_2 \alpha^2$ is much smaller than $\frac{\partial^2 L}{\partial \beta_1}(\theta) = \lambda_1 \alpha^2$. Property (iii) confirms this stability, proving that $\beta_2$ increases monotonically throughout the GD trajectory.

Now we are ready to characterize the sharpness along the GD trajectory, and demonstrate the existence of progressive sharpening and EoS in the following theorem.

**Theorem 4.3.** Let $T_1$ to be the first time such that $\lambda_1 \eta \alpha^2(T_1) \geq 1.5$ (Here $T_1$ can be $\infty$). Then for any $0 \leq t < T_1$, $\alpha > 0$ increases monotonically, and

$$\frac{1.1}{\eta} \leq S(\theta(t)) \leq \frac{1.7}{\eta}.$$

For $t \geq T_1$, we have

$$\frac{1.5}{\eta} \leq S(\theta(t)) \leq \frac{4.7}{\eta}.$$

Theorem 4.3 guarantees that the phenomenon of progressive sharpening before entering the EoS stage (Phase 1) and the sharpness bounded near $2/\eta$ during EoS (Phase 2). Specifically, when $t < T_1$, the sharpness keeps growing, as evidentiated by the monotonic increase of $\alpha$, yet stays below $2/\eta$. When $t \geq T_1$, the sharpness is bounded from below and above, near $2/\eta$, indicating the GD trajectory stays in the flat region and never escapes to a sharper region during EoS.

Furthermore, we prove the progressive sharpening (Phase 2.1) and self-stabilization (Phase 2.2) during EoS, when $\theta = (\alpha, \beta_1, \beta_2)$ is initialized within a more stable set $\widetilde{\mathcal{X}}(\eta)$:

$$\widetilde{\mathcal{X}}(\eta) := \{ (\alpha, \beta_1, \beta_2) \in \mathcal{X}(\eta) | \ \alpha \leq \sqrt{1.5/(\lambda_1 \eta)}, \ \beta_2 \leq 0.2/\alpha, \ \beta_1^2 \leq \frac{\lambda_2 \beta_2}{50\lambda_1 \alpha}(1 - \alpha\beta_2) \}.$$

Here $\widetilde{\mathcal{X}}(\eta)$ forms a significant subset of $\mathcal{X}(\eta)$. It represents a relatively flat region in $\mathcal{X}(\eta)$. Then we can derive the following theorem:

**Theorem 4.4** (Edge of Stability). Let $\theta(0) \in \widetilde{\mathcal{X}}(\eta)$. There exists $0 < T_2 < T_3$, such that:

• (Progressive Sharpening) For $t \in [0, T_2]$, we have

$$\alpha(t+1) > \alpha(t), \quad \text{and} \quad S(\theta(T_2)) > \frac{2.37}{\eta};$$

• (Self-stabilization) For $t \in (T_2, T_3]$, we have

$$\alpha(t+1) < \alpha(t), \quad \text{and} \quad S(\theta(T_3)) < \frac{2.01}{\eta}.$$

Here we demonstrate the monotonicity of $\alpha$. The monotonicity of $S(\theta)$ presents a more complex analytical challenge due to the absence of an explicit expression. Nevertheless, as demonstrated in Lemma 4.2, the magnitude of $S(\theta)$ is tightly bounded both above and below by $\alpha^2$. While we do not provide a strict proof of $S(\theta)$'s monotonicity, these bounds effectively characterize its behavior with considerable precision.

Theorem 4.4 gives a characterization of progressive sharpening and self-stabilization, which are consistent with the empirical observation in Damian et al. [2022] and Li et al. [2022b]. When $t < T_2$, sharpness tends to increase as $\alpha$ increases monotonically, and exceeds the stability threshold $2/\eta$. When $T_2 < t \leq T_3$, the sharpness drops, as $\alpha$ decrease monotonically. Additionally, we empirically demonstrate that during EoS (Figure 1), the GD dynamics repeatedly alternates between progressive sharpening (Phase 2.1) and self-stabilization (Phase 2.2).

### 4.2 Loss Decay Rate

In this section, we estimate a monotonic decay rate of the non-monotonic loss. Recall the loss function defined in (3.2):

$$L(\theta) = \underbrace{\frac{1}{2}\lambda_1(\alpha\beta_1)^2}_{L_1:\text{ oscillatory term}} + \underbrace{\frac{1}{2}\lambda_2(\alpha\beta_2 - 1)^2}_{L_2:\text{ convergence term}}. \tag{4.2}$$

The loss $L$ has two components, an oscillatory term $L_1$ and a convergence term $L_2$. The first term $L_1$ corresponds to learning the irrelevant feature $x_1$, and contains an oscillatory parameter $\beta_1$, as shown

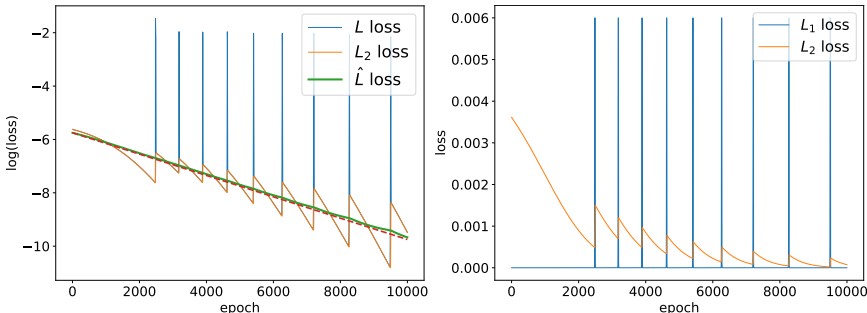

Figure 2: Same setting as Figure 1. In the left figure we plot $L_{(}\theta)$, $L_2(\theta)$ and $\widehat{L}(\theta)$ in log scale. We can see that $\widehat{L}(\theta)$ nicely reflects the decay rate for $L_2(\theta)$. The slope of the red dashed line is $2\log(1 - \frac{2\lambda_2}{\lambda_1})$, which nicely reflect the decrease rate of $\widehat{L}(\theta)$. In the right figure we plot $L_1(\theta)$ and $L_2(\theta)$. In most time $L_1(\theta)$ is near zero unless spikes occur.

in (ii) of Lemma 4.2. Empirically, $L_1$ exhibits drastic fluctuations, but rapidly drops to a level near zero every time the GD trajectory goes back to the stable region where $S(\theta) \leq 2/\eta$, as demonstrated in Figure 2. We thereby deduce that $L_1$ contributes to the spikes appearing in the loss dynamics, while $L_2$ dominates the loss descent.

Notably, $L_1$ barely influences the overall loss descent rate. This motivates us to focus on $L_2$ to estimate the monotonic decay rate. Our numerical experiments in Figure 2 indicate that $L_2$ well approximates the descent trend of $L$. However, $L_2$ is still non-monotonic, due to the fluctuations of $\alpha$. Thereby, we consider

$$\widehat{L}(\theta) = \frac{1}{2}\left(1 - \frac{\sqrt{2}\beta_2(t)}{\sqrt{\lambda_1\eta}}\right)^2,$$

by restricting $\alpha = \sqrt{2/(\lambda_1\eta)}$. We first show that $\widehat{L}$ is a good estimate for $L_2$:

**Lemma 4.5.** Let $\theta(0) \in \mathcal{X}(\eta)$. For any $0 \leq t \leq T_4$, where $T_4 := \min\{t \geq 0 : \beta_2(t) \geq 0.5\sqrt{\lambda_1\eta/2}\}$, we have $0.75L_2(\theta) \leq \widehat{L}(\theta) \leq 3.3L_2(\theta)$. Moreover, for any $0 \leq t \leq T_4$ such that $\alpha(t) \leq \sqrt{2/(\lambda_1\eta)}$, we have $0.75L_2(\theta) \leq \widehat{L}(\theta) \leq L_2(\theta)$.

Lemma 4.5 proves that $L_2(\theta)$ can be controlled by $\widehat{L}(\theta)$, especially when GD trajectory is in the stable region. Next, we present the decay rate for $\widehat{L}$:

**Theorem 4.6.** Let $\theta(0) \in \mathcal{X}(\eta)$. For any $0 \leq t \leq T_4$, with $T_4$ defined in Lemma 4.5, we have:

$$(1 - \frac{4\lambda_2}{\lambda_1})^2 \leq \frac{\widehat{L}(\theta(t+1))}{\widehat{L}(\theta(t))} \leq (1 - \frac{\lambda_2}{\lambda_1})^2.$$

Theorem 4.6 proves $\widehat{L}$ decays in a linear rate upper bounded by $(1 - \lambda_2/\lambda_1)^2$. The rate becomes faster as $\lambda_2/\lambda_1$ grows larger. This reflects how the relative scale between features influences the loss decay. As $\lambda_2/\lambda_1$ increases, the model becomes more sensitive to the relevant feature $x_2$, leading to faster loss descent. In addition, Figure 2 shows that the rate $(1 - 2\lambda_2/\lambda_1)^2$, which lies between the upper and lower bounds in Theorem 4.6, precisely estimates the decreasing speed of $L(\theta)$.

Our analysis primarily focuses on the interval $[0, T_4]$. While this formulation might appear constrained, it captures a substantial period of the optimization process, yielding valuable insights into the GD dynamics. Notably, Figure 2 demonstrates that the estimated decay rate maintains its validity well beyond the theoretically analyzed timeframe, suggesting broader applicability of our findings.

## 5  Connection with Existing Results

We present the connections of our theory to the gradient flow solution (GFS) proposed in Kreisler et al. [2023], and the "constrained trajectory" in Damian et al. [2022].

## 5.1 Connection with Gradient Flow Solution

Firstly, we introduce gradient flow (GF), which is the continuous form of GD: $\dot{\theta}(t) = -\nabla L(\theta(t))$. For any initialization $\theta$, define the gradient flow solution (**GFS**) $\mathcal{S}_{\mathrm{GF}}(\theta)$ as the limit of the GF trajectory, and denote $\phi(\theta)$ as the GFS sharpness, i.e. the loss sharpness at $\mathcal{S}_{\mathrm{GF}}(\theta)$.

Kreisler et al. [2023] study the GFS sharpness along the GD trajectory on scalar networks, proving its monotonic decrease. We extend this analysis to our two-layer neural network with two-dimensional input. Herein, we verify that GD reduces the GFS sharpness in this more general setting, while further elucidating the distinction between GFS sharpness and GD sharpness.

The following Lemma characterizes the GFS sharpness when the GF starts at $\theta(0) \in \mathcal{X}(\eta)$.

**Lemma 5.1.** Let $\eta \in [2/\lambda_1, \frac{1}{10\sqrt{\lambda_1\lambda_2}}]$. For any $\alpha(0), \beta(0) \in \mathcal{X}(\eta)$ satisfying $\alpha(0) = \beta_2(0)$, there exists $\beta_1(0) \in \mathbb{R}$ such that $\theta(0) = (\alpha(0), \beta_1(0), \beta_2(0)) \in \mathcal{X}(\eta)$. Moreover, the GF initialized at $\theta(0)$ converges to a solution with the following sharpness:

$$\phi(\theta(0)) \geq (1 - \frac{1}{4000})\lambda_1.$$

Lemma 5.1 reveals that when initialized from a subset of $\mathcal{X}(\eta)$, GF converges to sharp minima with sharpness lower bounded by $(1 - \frac{1}{4000})\lambda_1$. This result show that our analysis is beyond that in Kreisler et al. [2023], as they requires $\phi(0) \leq \frac{2\sqrt{2}}{\eta}$. In particular, given $\frac{1}{10\sqrt{\lambda_1\lambda_2}} \geq \frac{10}{\lambda_1}$, when $\eta \geq 5/\lambda_1$, the GFS sharpness $\phi(0) > 4/\eta$. While GD with large $\eta$ converges to flat minima whose sharpness is bounded by $2/\eta$ (Theorem 4.1), GF exhibits different trajectories and converges to solutions with greater sharpness.

Next we analyze GF initialized from points $\theta(t)$ along GD trajectory. The following theorem demonstrates that GD decreases the GFS sharpness, which is consistent with the observations in Kreisler et al. [2023].

**Theorem 5.2.** For any $t \geq 0$, we have

$$\phi(\theta(t)) \geq \frac{1 - \lambda_1\eta\beta_2^2(t)}{2\eta} + \frac{\lambda_1\sqrt{4 + (\frac{1}{\eta\lambda_1} - \beta_2^2(t))^2}}{2},$$

$$\phi(\theta(t)) \leq \frac{4.2 - \lambda_1\eta\beta_2^2(t)}{2\eta} + \frac{\lambda_1\sqrt{4 + (\frac{4.2}{\eta\lambda_1} - \beta_2^2(t))^2}}{2},$$

and both the upper and lower bounds monotonically decrease with time $t \geq 0$.

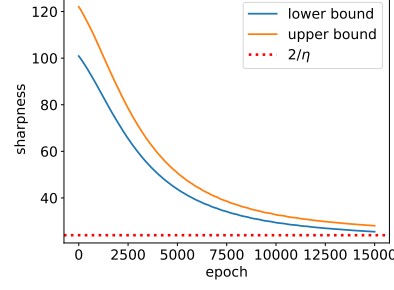

Figure 3: We choose $\eta = \frac{1}{12}$, and show the evolvement of upper and lower bound in Theorem 5.2

Here the monotonic decrease of the upper and lower bound in Theorem 5.2 is because of the monotonic increase of $\beta_2(t)$, as proved in Lemma 4.2 (iii). In Figure 3, we plot the upper and lower bound in Theorem 5.2. This demonstrates that both bounds do decrease monotonically with time, and implies a near-monotonic decrease of the GFS sharpness.

Figure 4 illustrates GF trajectories initiated from various points along the GD path. Recall that a greater $\alpha$ indicates higher sharpness, as shown in (i) of Lemma 4.2. The plot demonstrates how the GFSs transition from the highly sharp Minimizer 0 to Minimizer 3 with moderate sharpness $2/\eta$.

## 5.2 Connection with Constrained Trajectory

In this section, we draw connections to the "constrained trajectory" framework introduced by

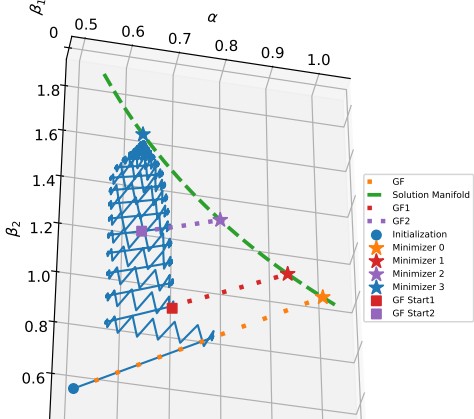

Figure 4: Same setting as Figure 1. We plot GFs starting from different points on the GD trajectory and the minimizers these GFs converge to.

Damian et al. [2022], who show that GD at the edge of stability implicitly follows projected gradient descent (PGD) under the constraint $S(\theta) \leq 2/\eta$. Specifically, they consider PGD on a so-called stable set $\mathcal{M}$, which is defined as

$$\mathcal{M}(\eta) := \left\{ \theta : S(\theta) \leq \frac{2}{\eta} \quad \text{and} \quad \nabla L(\theta) \cdot u(\theta) = 0 \right\},$$

where $u(\theta)$ is the eigenvector associated with the largest eigenvalue of Hessian $H(\theta)$. Then the PGD on $\mathcal{M}$ is formulated as

$$\theta_0^\dagger := \Pi_{\mathcal{M}}(\theta_0) \quad \text{and} \quad \theta_{t+1}^\dagger := \Pi_{\mathcal{M}}\left(\theta_t^\dagger - \eta \nabla L(\theta_t^\dagger)\right),$$

where the projection on $\mathcal{M}$ is defined as $\Pi_{\mathcal{M}}(\theta) := \arg\min_{\theta' \in \mathcal{M}} \|\theta - \theta'\|$. The trajectory of the PGD is referred to as "constrained trajectory".

As mentioned in Damian et al. [2022], the constrained trajectory is mainly determined by the sharpness condition, $S(\theta) \leq 2/\eta$, while the other condition $\nabla L(\theta) \cdot u(\theta) = 0$ of $\mathcal{M}$ is included to ensure the constrained trajectory stable, without affecting the stationary points of PGD on $\mathcal{M}$. Following the same principle, we choose our stable set as:

$$\mathcal{M}^\dagger(\eta) = \left\{ \theta : \frac{1}{\sqrt{\lambda_1 \eta}} \leq \alpha \leq \sqrt{\frac{2}{\lambda_1 \eta}}; \beta_1 = 0; 0 < \alpha\beta_2 \leq 9 \right\}.$$

The first condition $\alpha$ corresponds directly to maintaining sharpness near $2/\eta$ (Lemma 4.2). The second condition sets the oscillatory term $\beta_1$ to zero. Moreover, the third condition ensures parameter in $\mathcal{M}^\dagger(\eta)$ is not too far from solution manifold. The following lemma proves $\mathcal{M}^\dagger \subset \mathcal{M}$, and provides an explicit update rule for the PGD on $\mathcal{M}^\dagger$.

**Lemma 5.3.** Let $\eta \in [2/\lambda_1, 0.1]$ and $\theta(0) \in \mathcal{X}(\eta)$. Then $\mathcal{M}^\dagger(\eta)$ is a subset of $\mathcal{M}(\eta)$, and the PGD on $\mathcal{M}^\dagger(\eta)$ has the following update: $\beta_1^\dagger(t) \equiv 0$ and

$$\alpha^\dagger(t+1) = \mathbf{Clip}\left(\alpha^\dagger(t) - \eta \nabla_\alpha L(\theta^\dagger(t)), \sqrt{\frac{2}{\eta\lambda_1}}\right),$$

$$\beta_2^\dagger(t+1) = \beta_2^\dagger(t) - \eta \nabla_{\beta_2} L(\theta^\dagger(t)),$$

where $\mathbf{Clip}(x, c) = \text{sign}(x) \cdot \min\{|x|, c\}$.

Lemma 5.3 implies that the PGD on $\mathcal{M}^\dagger$ can be regarded as GD with $\beta_1$ restricted to zero and $\alpha$ clipped. We next characterize the PGD trajectory, derive its convergence rate during EoS, and demonstrate its value in understanding GD dynamics.

**Theorem 5.4.** Let $\tilde{t} := \min\{t \geq 0 : \alpha^\dagger(\tilde{t}) = \sqrt{2/(\eta\lambda_1)}\}$. For any $t$ such that $0 \leq t < \tilde{t}$, the PGD on $\mathcal{M}^\dagger$ satisfies

$$\alpha^\dagger(t+1) > \alpha^\dagger(t) \quad \text{and} \quad L(\theta^\dagger(t+1)) < L(\theta^\dagger(t)).$$

For any $t \geq \tilde{t}$, we have

$$L(\theta^\dagger(t+1)) = \left(1 - \frac{2\lambda_2}{\lambda_1}\right)^2 L(\theta^\dagger(t)) \quad (5.1)$$

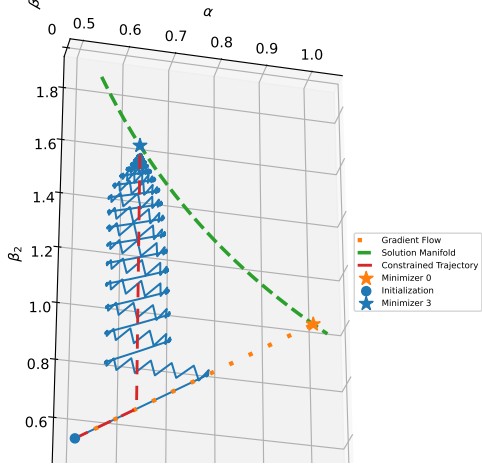

Figure 5: Same stetting as Figure 1. We plot GD trajectory with learning rate $\eta = \frac{1}{20}$, as well as the GF and constrained trajectory starting from the same initialization.

Theorem 5.4 shows the loss monotonically decreases along the constrained trajectory, and provides an explicit convergence rate. We can see that this rate lies between the upper bound and lower bound in Theorem 4.6, and in Figure 2 we can see that (5.1) predicts the decay rate of $L$ precisely.

The constrained trajectory analysis complements our approach in Section 4.2 in an interesting way. Both methods aim to understand the core convergence behavior by handling the oscillatory dynamics, but through different means. The constrained trajectory directly enforces stability by projecting $\theta$

onto a stable set where $\beta_1 = 0$, while our previous analysis in Section 4.2 studies the convergence through $L_2$ (defined in (4.2)), while allowing the oscillatory parameter $\beta_1$ to vary naturally. Despite these distinct approaches, both analyses arrive at similar convergence rates, indicating that $\beta_1$ barely impacts the loss descent and reinforcing our understanding of the fundamental dynamics at the EoS.

We also visualize the constrained trajectory in Figure 5. We show that GD first "sharpens" to the unstable region where $\alpha$ is large, and then "self-stabilize" to the stable region. After that, another sharpening starts and the trajectory enters the next cyclical behavior. As illustrated in Figure 5, during EoS stage, GD trajectory will follow the constrained trajectory rather that GF trajectory.

## 6 Discussion

### 6.1 Alignment with the EoS Phenomena in Practice

To our best knowledge, the phenomenon in our setting, including the periodic progressive sharpening and self-stabilization, and the periodic loss spikes, have not been observed in previous minimalist settings [Zhu et al., 2022, Kreisler et al., 2023, Wang et al., 2023, Chen and Bruna, 2023]. Our setting is based on the two dimensional input data with two features in different scale. This is different from most previous minimalist settings such as Zhu et al. [2022], Kreisler et al. [2023] where they suppose a scalar input. Recall that in our setting, we suppose a large-scale but irrelevant feature and a small scale and relevant feature. Such construction is the key for our phenomenon, as the model has the trend to progressive sharpening (to increase its norm) for fitting the small scale feature, but the large-scale irrelevant feature (or noise) will provide a regularization for the model, which provides a self-stabilization effect.

We argue that observations in our settings are more similar to practical results Cohen et al. [2021], Damian et al. [2022] than observation in Zhu et al. [2022], Kreisler et al. [2023]. In experiments from Cohen et al. [2021], Damian et al. [2022], it's easy to see the periodic progressive sharpening, self-stabilization and the loss spikes, see Figure 6 and 7. These phenomenon are also observed in our synthetic setting. In contrast, in the scalar network setting in Zhu et al. [2022], Kreisler et al. [2023], the sharpness jump up and down around $\frac{2}{\eta}$, and their loss function will monotonically decrease after first a few steps. See Figure 8 and Figure 9 for the experiments on scalar networks.

### 6.2 Trade-off in Large Learning Rates

An interesting trade-off emerges in our setting: larger learning rates tend to drive the optimization toward flatter minima, yet such minima are often located farther from the initialization. More specifically, since $\alpha$ remains close to $\sqrt{\frac{2}{\lambda_1 \eta}}$ at both initialization and convergence, while $\beta_1$ starts from a small value and converges to 0, the distance between initialization and the final minimal is primarily determined by $\beta_2$, and its scale is approximately on the order of $O(\frac{1}{\alpha}) = O(\sqrt{\lambda_1 \eta})$. Thus a larger learning rate $\eta$ results in a greater distance between the initialization and the final minimum.

This introduces a tension between the step size and the distance to convergence, which implies that a larger learning rate does not necessarily lead to faster convergence. While this conclusion is consistent with previous observations in the literature, our work highlights a distinct underlying mechanism. In convex settings, where the minimum is unique, slower convergence under large step sizes is typically attributed to overshooting. In non-convex settings, although minima are not unique, prior work generally focuses on the per-step decrease in objective value, rather than the specific minimum reached.

In contrast, our analysis explicitly shows that larger learning rates converge to flatter—and thus more distant—minima. This perspective reveals a new form of trade-off: the increase in step size is counteracted by the increased distance to the final solution, providing a novel explanation for the non-monotonic relationship between learning rate and convergence speed. We believe this insight offers a fresh angle for understanding the dynamics of optimization in non-convex landscapes, and opens the door to future work on characterizing such trade-offs in more general settings.

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

## A    Phenomena in other works

We begin by examining an experiment from Cohen et al. [2021], in which they train a two-layer tanh network to approximate a Chebyshev polynomial. The observed phenomenon closely resembles our findings. The sharpness initially increases, then rapidly decreases to a low level, before growing again and entering a cyclic pattern. Large spikes occur and quickly subside during this process. Figure 6 illustrates their results.

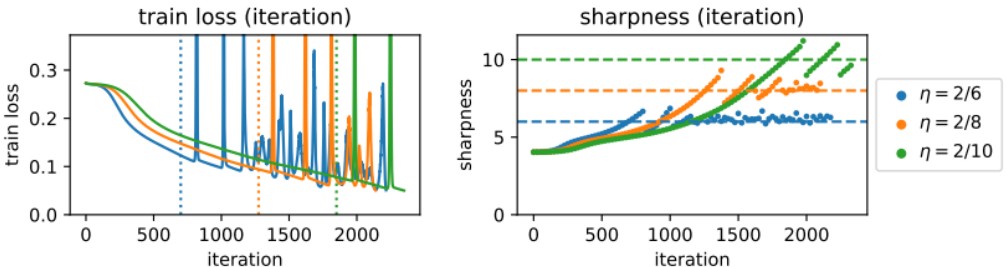

Figure 6: Experiment from Cohen et al. [2021]. A two-layer tanh network is trained to approximate a Chebyshev polynomial using Gradient Descent (GD) with learning rates $\eta = \frac{2}{6}, \frac{2}{8}$, and $\frac{2}{10}$.

Similar observations are reported in Li et al. [2022b] and Damian et al. [2022]. For instance, Damian et al. [2022] train a transformer with MSE Loss on the SST2 dataset. We present their results in Figure 7.

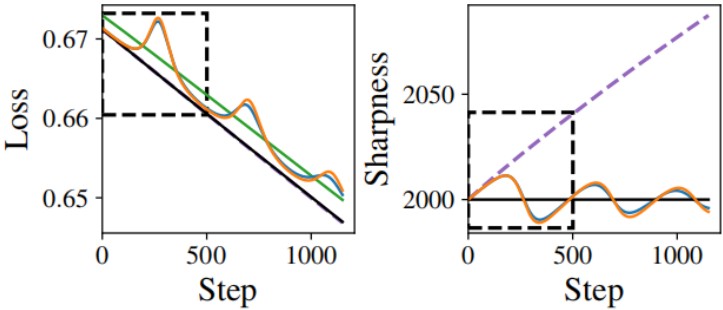

Figure 7: Experiment from Damian et al. [2022]. A transformer model is trained on the Stanford Sentiment Treebank 2 (SST-2) dataset using Mean Squared Error (MSE) as the loss function.

These experiments from various studies align closely with our numerical experiments. However, we note a discrepancy between these practical experiments and those presented in Zhu et al. [2022] and Kreisler et al. [2023]. In their experiments on scalar networks, the loss oscillates during the initial steps but subsequently decreases monotonically. The sharpness does not exhibit a progressive sharpening process; instead, it fluctuates around $2/\eta$ from the initialization. Figures 8 and 9 provide detailed illustrations of these observations. Based on these findings, we posit that the Edge of Stability (EoS) phenomenon differs from that described in Zhu et al. [2022] and Kreisler et al. [2023].

## B    More Experiments and Details

### B.1    Visualization of Practical Model Parameters in 3D Space

We visualize part of the GD trajectory with different learning rates, as well as the Gradient Flow (GF) trajectory, while training a four-layer MLP on a binary classification problem using the CIFAR-10 dataset. The GD trajectory closely follows the GF trajectory before entering the EoS regime. Subsequently, the GD trajectory enters an unstable region, beginning to oscillate, and its long-term movement shifts to a new direction that differs from the GF direction. This behavior is similar to GD's behavior observed in our results presented in Figure 5. Additionally, we observe that GD with

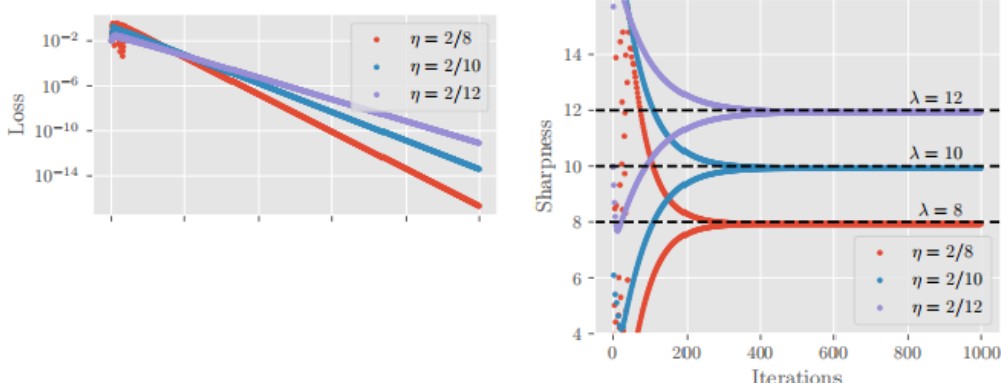

Figure 8: Experiment from Zhu et al. [2022]. Gradient Descent (GD) is applied to the loss function $(1 - x^2y^2)^2$, which represents a special case of a scalar network setting. The learning rates $\eta$ are chosen to be $\frac{2}{8}, \frac{2}{10}$, and $\frac{2}{12}$.

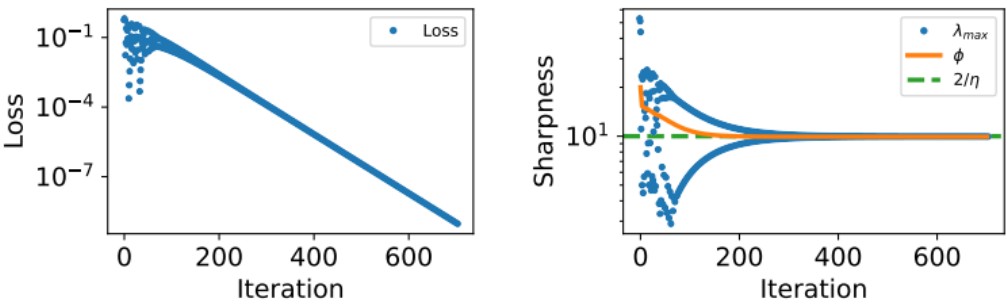

Figure 9: Experiment from Kreisler et al. [2023]. A depth-4 scalar network is trained using Gradient Descent (GD) for $10^4$ steps with a learning rate $\eta = 0.2$. The loss function is defined as $(1 - xyzw)^2$.

smaller learning rates follows the GF for a longer distance before beginning to change direction and oscillate in a new region.

## C   Extension of the Setting

Here are some extensions of our setting. First, by exploiting the rotational invariance of Gradient Descent in neural networks, we can assume that the input data $x \in \mathbb{R}^d$ satisfies

$$x = x_1v_1 + x_2v_2,$$

where $v_1, v_2 \in \mathbb{R}^d$, $\|v_1\| = \|v_2\| = 1$, and $\langle v_1, v_2 \rangle = 0$. The pair $(x_1, x_2)$ and the target function $f^* : \mathbb{R}^d \to \mathbb{R}$ follow the same assumptions as in our original setting:

$$\begin{pmatrix} x_1 \\ x_2 \end{pmatrix} \sim \mathcal{N}\left(0, \begin{pmatrix} \lambda_1 & 0 \\ 0 & \lambda_2 \end{pmatrix}\right), \quad \text{and } f^*(x) = x_2.$$

Then, the covariance matrix of the input $x$ is given by

$$\mathbb{E}[xx^T] = \lambda_1 v_1 v_1^T + \lambda_2 v_2 v_2^T.$$

This covariance can be a non-diagonal matrix in $\mathbb{R}^{d \times d}$, indicating that our setting naturally extends to high-dimensional and non-diagonal data distributions.

In addition, the current setting can be extended to multiple features. Specifically, we can extend the input from $(x_1, x_2)$ to $(x_1, x_2, x_3)$, where $x_3$ is independent of both $x_1$ and $x_2$, and has the same

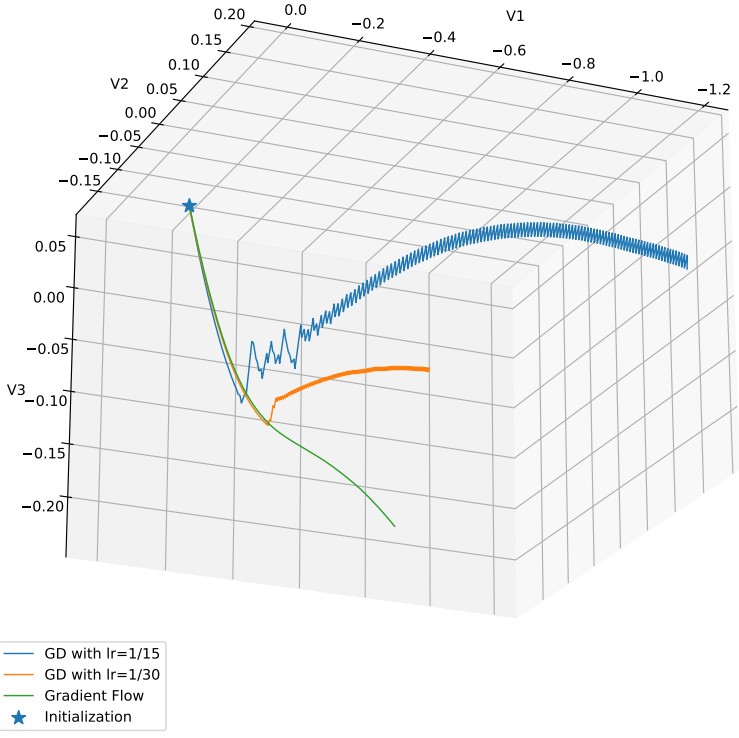

Figure 10: Visualization of GD trajectory for a four layers MLP training on Cifar10

distribution as $x_2$. We redefine the target as $y = x_2 + x_3$, and correspondingly extend the parameter vector from $(\alpha, \beta_1, \beta_2)$ to $(\alpha, \beta_1, \beta_2, \beta_3)$. The loss function then becomes

$$\frac{1}{2}\lambda_1(\alpha\beta_1)^2 + \frac{1}{2}\lambda_2(\alpha\beta_2 - 1)^2 + \frac{1}{2}\lambda_2(\alpha\beta_3 - 1)^2.$$

By symmetry, if the initialization satisfies $\beta_2(0) = \beta_3(0)$, then $\beta_2(t) = \beta_3(t)$ for all $t$. Under this condition, the loss simplifies to

$$\frac{1}{2}\lambda_1(\alpha\beta_1)^2 + \frac{1}{2}(2\lambda_2)(\alpha\beta_2 - 1)^2,$$

which corresponds exactly to the original setting with $\lambda_1' = \lambda_1$ and $\lambda_2' = 2\lambda_2$, and the training dynamics remain equivalent.

Although this construction may appear somewhat artificial, it demonstrates that introducing an additional independent variable $x_3$ with a small scale has a similar effect on the learning dynamics as increasing the scale of $x_2$. If the initialization does not satisfy $\beta_2(0) = \beta_3(0)$, the analysis becomes more complex, but the overall behavior remains closely aligned with the original setting.

## D    Theoretical Properties of Initialization

Our initialization allows us to explore the GD dynamics across a wide range of learning rates, all starting from the same initial point. This stands in stark contrast to previous minimalist analyses such as Zhu et al. [2022] and Chen and Bruna [2023], where their initializations permit only a narrow selection of learning rates.

To see this, we define a special set $\mathbb{Y} \subset \mathbb{R}^3$ that is independent of the learning rate:

$$2\sqrt{5} \cdot \left(\frac{\lambda_2}{\lambda_1}\right)^{\frac{1}{4}} \le \alpha < 1, \quad \max\left\{\alpha, \frac{\sqrt{3}}{10\alpha}\right\} \le \beta_2 < \frac{1}{\alpha}, \quad \frac{\lambda_2\beta_2}{500\lambda_1\alpha} \le \frac{\beta_1^2}{1 - \alpha\beta_2} \le \frac{\lambda_2\beta_2}{\lambda_1\alpha}.$$

We then have the following proposition:

**Proposition D.1.** For any $\theta \in \mathbb{Y}$, there exists a learning rate $\eta \leq \frac{1}{10\sqrt{\lambda_1\lambda_2}}$ such that for any decay rate $r \in [0.55, 1]$, $\theta \in \mathcal{X}(r\eta)$.

*Proof.* Since $\lambda_1\alpha^2 \geq 20\sqrt{\lambda_1\lambda_2}$, we know that there exists a learning rate $\eta$ such that $\lambda_1\alpha^2\eta = 2$. First, we show that $\theta \in \mathcal{X}(\eta)$:

(i) $\lambda_1\alpha^2\eta = 2$ and $\alpha > 0$, so the condition $\sqrt{\frac{1.1}{\lambda_1\eta}} \leq \alpha \leq \sqrt{\frac{2}{\lambda_1\eta}}$ is satisfied.

(ii) $\alpha \leq \beta_2 < \frac{1}{\alpha}$ is satisfied by the definition of $\mathbb{Y}$. Since $\beta_2 \geq \frac{\sqrt{3}}{10\alpha}$, we know $\beta_2 \geq \frac{\sqrt{3}}{20\alpha}$. Since $\eta = \frac{2}{\lambda_1\alpha^2}$, we have $\frac{\sqrt{6\eta\lambda_1}}{20} = \frac{\sqrt{3}}{10\alpha} \leq \beta_2$. So the condition $\max\left\{\frac{\sqrt{6\eta\lambda_1}}{20}, \frac{3}{20\alpha}, \alpha\right\} \leq \beta_2 < \frac{1}{\alpha}$ is satisfied.

(iii) The condition for $\beta_1$ is naturally satisfied by the definition of $\mathbb{Y}$.

Next, note the following property that for $r \in [0.55, 1]$, we have:

$$1.1 \leq \lambda_1\alpha^2 r\eta \leq 2, \quad \frac{\sqrt{6r\eta\lambda_1}}{20} \leq \frac{\sqrt{6\eta\lambda_1}}{20}.$$

Therefore, it is apparent that if $\theta \in \mathcal{X}(\eta)$, then for $r \in [0.55, 1]$, $\theta \in \mathcal{X}(r\eta)$. $\quad\square$

# E   Proofs for Section 4

## E.1   Proof of Lemma 4.2 and Theorem 4.3

To simplify the proof, we assume $K$, which is the absolute constant in our setting for the lower bound of $\frac{\lambda_1}{\lambda_2}$, satisfying $K \geq 10^4$. First we prove the following Proposition:

**Proposition E.1.** For any $t \geq 0$, let $\widehat{t}$ to be the first time such that $\lambda_1\eta\alpha^2(\widehat{t}) \geq 1.5$, then we have:

$$1.5 \leq \lambda_1\eta\alpha^2(t) \leq 4.19 \quad \text{for } t \geq \widehat{t}. \tag{E.1}$$

and $\lambda_1\alpha^2(t)$ keeps to increase before $\widehat{t}$ ($\widehat{t}$ can be $\infty$).

*Proof.* Note that we use learning rate $\eta \leq 0.1$. The update equation can be written as:

$$\alpha(t+1) = \alpha(t) - \eta\left(\lambda_1\beta_1^2(t)\alpha(t) - \lambda_2\beta_2(t)(1 - \alpha(t)\beta_2(t))\right), \tag{E.2}$$

$$\beta_1(t+1) = \mathbf{Clip}\left(\beta_1(t) - \eta\lambda_1\alpha^2(t)\beta_1(t), \frac{1}{2} \cdot (\frac{\lambda_2}{\lambda_1})^{\frac{1}{4}}\right), \tag{E.3}$$

$$\beta_2(t+1) = \beta_2(t) + \eta\lambda_2\alpha(t)(1 - \alpha(t)\beta_2(t)), \tag{E.4}$$

where $\mathbf{Clip}(x, c) = \mathrm{sign}(x) \cdot \min\{|x|, c\}$. We can get that:

$$(1 - \alpha(t+1)\beta_2(t+1)) < (1 - \alpha(t)\beta_2(t)) \cdot$$
$$\left[1 - \eta\lambda_2\alpha^2(t) - \eta\lambda_2\beta_2^2(t) - \eta^2\lambda_2^2\alpha(t)\beta_2(t)\left(1 - \alpha(t)\beta_2(t)\right)\right]. \tag{E.5}$$

Note that the following are the initialization conditions:

$$1.1 \leq \eta\lambda_1\alpha^2(0) < 2, \quad \max\{\frac{\sqrt{6\eta\lambda_1}}{20}, \frac{3}{20\alpha(0)}, \alpha(0)\} \leq \beta_2(0) < \frac{1}{\alpha(0)}, \tag{E.6}$$

$$\frac{\lambda_2\beta_2(0)}{500\lambda_1\alpha(0)}\left(1 - \alpha(0)\beta_2(0)\right) \leq \beta_1^2(0) \leq \frac{\lambda_2\beta_2(0)}{\lambda_1\alpha(0)}\left(1 - \alpha(0)\beta_2(0)\right). \tag{E.7}$$

We set $t^\dagger$ to be the largest time such that for any $t \leq t^\dagger$ we have $\alpha > 0$ and:

$$\eta\lambda_2\alpha^2(t) + \eta\lambda_2\beta_2^2(t) \leq 0.01.$$

We will first prove Proposition E.1 for $t \leq t^\dagger + 1$. For $t \leq t^\dagger + 1$ we know that in (E.5) we have:

$$1 - \eta\lambda_2\alpha^2(t) - \eta\lambda_2\beta_2^2(t) - \eta^2\lambda_2^2\alpha(t)\beta_2(t)\left(1 - \alpha(t)\beta_2(t)\right) \geq 1 - 0.01 - \frac{\eta^2\lambda_2^2}{4} > 0.$$

Since $\alpha(0)\beta_2(0) < 1$, based on (E.5) we can see that $\alpha(t)\beta_2(t) < 1$ for $t \leq t^\dagger + 1$. Therefore, if Proposition E.1 holds for $t \leq t^\dagger + 1$, we will get $\alpha(t) > 0$ for $t \leq t^\dagger + 1$, and we have:

$$\eta\lambda_2\alpha^2(t) + \eta\lambda_2\beta_2^2(t) \leq \frac{\lambda_2\lambda_1\eta\alpha^2(t)}{\lambda_1} + \frac{\eta^2\lambda_1\lambda_2}{\lambda_1\eta\alpha^2(t)} \leq \frac{5\lambda_2}{\lambda_1} + \frac{1}{1.1 \cdot 10^2} \leq 0.01,$$

which means $t^\dagger = t^\dagger + 1$, thus $t^\dagger = \infty$. Therefore, it suffices to prove Proposition E.1 for $t \leq t^\dagger + 1$, which yields $t^\dagger = \infty$ and $\alpha(t)\beta_2(t) < 1$ for all $t$.

Let $t_0$ be the last time that $\lambda_1\eta\alpha^2(t_0) < 2$, with $t_0 \leq \infty$. If $\alpha^2$ does not increase monotonically before $t_0$, let $t_1$ be the first time such that $\alpha^2(t_1 + 1) < \alpha^2(t_1)$, where $t_1 \geq 1$. We have that for any $t < t_1$, $\alpha(t)\beta_2(t) < 1$ and:

$$\left(1 - 0.01 - \frac{1}{4}\eta^2\lambda_2^2\right)(1 - \alpha(t)\beta_2(t)) \leq 1 - \alpha(t+1)\beta_2(t+1) \leq 0.85,$$

$$\alpha(t+1) \leq \left(1 + \frac{\eta\lambda_2}{4\alpha^2(t)}\right)\alpha(t) = \left(1 + \frac{\eta^2\lambda_1\lambda_2}{4\lambda_1\eta\alpha^2(t)}\right)\alpha(t) \leq \left(1 + \frac{1}{400\lambda_1\eta\alpha^2(t)}\right)\alpha(t).$$

When $t = t_1 - 1$, we have:

$$(1 - \eta\lambda_1\alpha^2(t))^2 \cdot \left(1 + \frac{\eta^2}{4\lambda_1\eta\alpha^2(t)}\right) > 1 - 0.01 - \frac{1}{4}\eta^2\lambda_2^2 > 0.98.$$

Thus, for $t = t_1 - 1$:

$$\lambda_1\eta\alpha^2(t) \geq 1.95.$$

Moreover, for $t = t_1 - 1$, we have:

$$\beta_1^2(t) \leq \frac{\lambda_2}{4\lambda_1\alpha^2(t)} \leq \frac{\lambda_2\eta}{7.8}.$$

Before $t_0$, $\beta_1^2$ decreases monotonically, and when $\eta\lambda_1\alpha^2 \leq 1.9$, $\beta_1^2$ decreases with an exponential rate of at least $0.81$. Using Bernoulli's inequality, we have:

$$\inf_{t_1 \leq t \leq t_0}\{\eta\lambda_1\alpha^2\} \geq 1.9 \cdot \left(1 - \frac{5\lambda_2\eta}{7.8}\right)^2 > 1.5.$$

Let $\hat{t}$ be the first time $\eta\lambda_1\alpha^2(t) \geq 1.5$. Then $\alpha^2(t)$ monotonically increases before $\hat{t}$ and $\inf_{\hat{t} \leq t \leq t_0}\{\eta\lambda_1\alpha^2(t)\} \geq 1.5$. This also holds if $\alpha^2(t)$ increases monotonically before $t_0$. Next, we suppose $t_0 < \infty$ to analyze what will happen if GD enters the unstable regime.

Let $t_2$ be the first time after $t_0$ such that $\eta\lambda_1\alpha^2(t_2) < 2$, where $t_2 \leq \infty$. If $t_2 = \infty$, it is apparent that $\lim_{t \to \infty}\lambda_1\alpha^2(t) = 2$. Therefore, there will exist $t_1 < \infty$, such that

$$\lambda_1\alpha^2(t_1 + 1) = \sup_{t_0 < t \leq t_2}\{\eta\lambda_1\alpha^2(t)\}.$$

For $t_0 < t \leq t_2$, we have:

$$\lambda_1\left(\frac{1}{2} \cdot \left(\frac{\lambda_2}{\lambda_1}\right)^{\frac{1}{4}}\right)^2\alpha^2(t) \geq \frac{\sqrt{\lambda_2}}{2\eta\sqrt{\lambda_1}} \geq \frac{\lambda_2}{4} > \lambda_2\alpha(t)\beta_2(t)(1 - \alpha(t)\beta_2(t)).$$

We can deduce that for $t_0 < t \leq t_1$, $|\beta_1(t)| < \frac{1}{2} \cdot \left(\frac{\lambda_2}{\lambda_1}\right)^{\frac{1}{4}}$. Thus, we know that before $t_1$, the update equation is the same as GD without any constraints.

We set undefined coefficients $n_1$, $n_2$ and an undetermined time $t^\star$ such that:

$$\lambda_1\eta\left(\alpha^2(t+1) - \alpha^2(t)\right) \geq \frac{1}{n_1} \quad \text{for} \quad t^\star \leq t \leq t_0,$$

$$\lambda_1\eta\left(\alpha^2(t+1) - \alpha^2(t)\right) \leq \frac{1}{n_2} \quad \text{for} \quad t_0 < t \leq t_2,$$

where $n_1 > n_2$. We will estimate $n_1$, $n_2$, and $\frac{n_1}{n_2}$ later. Then we will have: $\lambda_1\eta\alpha^2(t^\star + t) \geq 1.1 + \frac{t}{n_1}$ and $\lambda_1\eta\alpha^2(t^\star + \frac{9n_1}{10}) \geq 2$. Then we have:

$$\left|\prod_{i=t^\star}^{t_0}(\lambda_1\eta\alpha^2(i) - 1)\right| \geq \prod_{i=0}^{\frac{9n_1}{10}}\left(\frac{\frac{n_1}{10} + i}{n_1}\right) = \frac{n_1!}{(\frac{n_1}{10})!n_1^{\frac{9n_1}{10}}} \geq \frac{\sqrt{2\pi n_1}(\frac{n_1}{e})^{n_1}}{e^{\frac{5}{6n_1}}\sqrt{\pi n_1}(\frac{n_1}{10e})^{\frac{n_1}{10}}n_1^{\frac{9n_1}{10}}} = \left(\frac{10^{\frac{1}{9}}}{e}\right)^{\frac{9n_1}{10}} \cdot \frac{\sqrt{2}}{e^{\frac{5}{6n_1}}}.$$

We suppose that $\lambda_1\eta\alpha^2(t_1) = \lambda + 1$ where $\lambda > 1$. Then we will have $\lambda_1\eta\alpha^2(t) - 1 \geq \lambda - \frac{t}{n_2}$ and $\lambda_1\eta\alpha^2((\lambda-1)n_2) - 1 \geq 1$. So we have:

$$\left|\prod_{i=t_0+1}^{t_1} (\lambda_1\eta\alpha^2(i) - 1)\right| \geq \frac{(\lambda n_2)!}{n_2! n_2^{(\lambda-1)n_2}} \geq \left(\frac{\lambda^\lambda}{e^{\lambda-1}}\right)^{n_2} \cdot \frac{\sqrt{\lambda}}{e^{\frac{1}{12n_2}}}.$$

Therefore, we can obtain:

$$\left|\prod_{i=t^\star}^{t_1} (\lambda_1\eta\alpha^2(i) - 1)\right| \geq \left(\frac{10^{\frac{1}{9}}}{e}\right)^{\frac{9n_1}{10}} \cdot \left(\frac{\lambda^\lambda}{e^{\lambda-1}}\right)^{n_2} \cdot \frac{\sqrt{2\lambda}}{e^{\frac{10n_2+n_1}{12n_1 n_2}}}.$$

Since $\alpha(t+1) \leq \alpha(t) + \lambda_2\eta\beta_2(t)(1 - \alpha(t)\beta_2(t))$, we have:

$$\begin{aligned}
\lambda_1\eta\alpha^2(t+1) &\leq \lambda_1\eta(\alpha(t) + \eta\lambda_2\beta_2(t)(1 - \alpha(t)\beta_2(t)))^2 \\
&= \lambda_1\eta\alpha^2(t) + 2\eta^2\lambda_1\lambda_2\alpha(t)\beta_2(t)(1 - \alpha(t)\beta_2(t)) + \eta^3\lambda_1\lambda_2^2\beta_2^2(t)(1 - \alpha(t)\beta_2(t))^2 \\
&\leq \lambda_1\eta\alpha^2(t) + \frac{\lambda_1\lambda_2\eta^2}{2} + \frac{\lambda_1^2\lambda_2^2\eta^4}{16\lambda_1\eta\alpha^2(t)} \leq \lambda_1\eta\alpha^2(t) + \frac{1}{200} + \frac{1}{1600}.
\end{aligned}$$

Given that $\eta \leq 0.1$, we can set $n_1$ and $n_2$ such that $\min\{n_1, n_2\} \geq 100$. Since we know:

$$\min_{t_0 \leq t \leq t_2}\{\alpha(t)\beta_2(t)\} \geq \max_{t^\star \leq t \leq t_0}\{\alpha(t)\beta_2(t)\} \geq 0.15,$$

we can derive:

$$\frac{\min_{t^\star \leq t \leq t_0} \alpha(t)\beta_2(t)(1 - \alpha(t)\beta_2(t))}{\max_{t_0 \leq t \leq T} \alpha(t)\beta_2(t)(1 - \alpha(t)\beta_2(t))} \geq \frac{1}{2}.$$

If $\lambda_1\alpha(t_0)\beta_1^2(t_0) \leq \frac{1}{50}\lambda_2\beta_2(t_0)(1 - \alpha(t_0)\beta_2(t_0))$, suppose $t^\star$ is the largest time before $t_0$ such that:

$$\lambda_1\alpha(t)\beta_1^2(t) > \frac{1}{50} \cdot \lambda_2\beta_2(t)(1 - \alpha(t)\beta_2(t)) \quad \text{for} \quad t = t^\star - 1.$$

Since before $t_0$, we have $\beta_1^2(t+1) \geq (1.1-1)^2\beta_1^2(t)$, we will have:

$$\frac{1}{5000}\lambda_2\beta_2(t^\star)(1 - \alpha(t^\star)\beta_2(t^\star)) \leq \lambda_1\alpha(t^\star)\beta_1^2(t^\star) \leq \frac{1}{50}\lambda_2\beta_2(t^\star)(1 - \alpha(t^\star)\beta_2(t^\star)).$$

Based on the definition of $t^\star$, we obtain:

$$\frac{\min_{t^\star \leq t \leq t_0}(\alpha^2(t+1) - \alpha^2(t))}{\max_{t_0 < t \leq T}(\alpha^2(t+1) - \alpha^2(t))} \geq \frac{1}{2.04}.$$

In conclusion, we can set $n_1$ and $n_2$ such that $\frac{n_1}{n_2} \leq 2.04$.

Combining all the above equations, we have:

$$\left(\frac{10^{1/9}}{e}\right)^{9n_1/10} \cdot \left(\frac{\lambda^\lambda}{e^{\lambda-1}}\right)^{n_2} \cdot \frac{\sqrt{2\lambda}}{e^{(10n_2+n_1)/(12n_1 n_2)}} > 100 \quad \text{when} \quad \lambda = 3.18.$$

Since $\alpha(t_1) \geq \alpha(t^\star)$ and $\alpha(t^\star)\beta_2(t^\star)(1 - \alpha(t^\star)\beta_2(t^\star)) \geq \frac{1}{2}\alpha(t_1)\beta_2(t_1)(1 - \alpha(t_1)\beta_2(t_1))$, we have:

$$\begin{aligned}
\lambda_1\eta\beta_1^2(t_1)\alpha^2(t_1) &> 10^4 \cdot \lambda_1\eta\beta_1^2(t^\star)\alpha^2(t^\star) \\
&> \frac{10^4\eta}{5 \cdot 10^3} \cdot \lambda_2\alpha(t^\star)\beta_2(t^\star)(1 - \alpha(t^\star)\beta_2(t^\star)) \\
&\geq \lambda_2\eta\alpha(t_1)\beta_2(t_1)(1 - \alpha(t_1)\beta_2(t_1)).
\end{aligned}$$

If $\lambda_1\alpha(t_0)\beta_1^2(t_0) > \frac{1}{50} \cdot \lambda_2\beta_2(t_0)(1 - \alpha(t_0)\beta_2(t_0))$, then we can see:

$$\begin{aligned}
\lambda_1\eta\beta_1^2(t_1)\alpha^2(t_1) &> 10^4 \cdot \lambda_1\eta\beta_1^2(t_0)\alpha^2(t_0) \\
&> \frac{10^4\eta}{5 \cdot 10^3} \cdot \lambda_2\alpha(t_0)\beta_2(t_0)(1 - \alpha(t_0)\beta_2(t_0)) \\
&> \lambda_2\eta\alpha(t_1)\beta_2(t_1)(1 - \alpha(t_1)\beta_2(t_1)).
\end{aligned}$$

Therefore, we have $\alpha(t_1 + 1) < \alpha(t_1)$, which contradicts the definition of $t_1$. So we can obtain $\lambda < 3.19$. Consequently, we have proved that:

$$\sup_{t_0 < t \leq t_2} \{\eta\lambda_1\alpha^2(t)\} = \lambda_1\eta\alpha^2(t_1 + 1) \leq \lambda_1\eta\alpha^2(t_1) + 0.01 \leq 4.19 < 4.2.$$

Next, we suppose $t_2 < \infty$ to examine the consequences. Using the property of the constraint on $\beta_1$ with constant $c = \frac{1}{2}(\frac{\lambda_2}{\lambda_1})^{\frac{1}{4}}$, we can obtain a lower bound for $\alpha^2(t)$ as:

$$\lambda_1\eta\alpha^2(t + 1) \geq (1 - \lambda_1\eta c^2)^2 \cdot \lambda_1\eta\alpha^2(t).$$

Let $t_3$ be the first time such that $\lambda_1\eta\alpha^2(t) < 1.7$. It is easy to see that:

$$\lambda_1\eta\alpha^2(t_3) \geq 1.7 \cdot (1 - \lambda_1\eta c^2)^2.$$

Evidently, for $t \geq t_3$, we have:

$$\beta_1^2(t + 1) \leq \frac{1}{2}\beta_1^2(t + 1).$$

Therefore, by using Bernoulli's inequality, we obtain that for $t \geq t_5$:

$$\lambda_1\eta\alpha^2(t) \geq 1.7 \cdot (1 - 2\lambda_1\eta c^2)^2.$$

Using the condition that $2\lambda_1\eta c^2 \leq \frac{\eta\sqrt{\lambda_1\lambda_2}}{2} \leq \frac{1}{20}$, we get $100\eta\alpha^2(t) \geq 1.5$ for $t \geq t_5$. Then we know $\beta_1^2(t + 1) \geq \frac{1}{4}\beta_1^2(t)$ for $t \geq t_3$.

Let $t_4$ be the first time after $t_3$ such that $\alpha^2(t)$ starts to increase again. We will discuss whether $\beta_1^2(t_4)$ will satisfy the initialization condition. The left side is immediately due to the definition of $t_4$. Let $\widehat{t_4} = t_4 - 1$. Since we have:

$$\beta_2(\widehat{t_4})(1 - \alpha(\widehat{t_4})\beta_2(\widehat{t_4})) \geq (0.09 - \frac{\eta^2\lambda_2^2}{4})\beta_2(t_4)(1 - \alpha(t_4)\beta_2(t_4))$$
$$\geq 5^{-3} \cdot \beta_2(t_4)(1 - \alpha(t_4)\beta_2(t_4)),$$

the right side of the initial condition is simply due to:

$$\beta_1^2(t_4) \geq \frac{1}{4}\beta_1^2(\widehat{t_4})$$
$$\geq \frac{\lambda_2\beta_2(\widehat{t_4})(1 - \alpha(\widehat{t_4})\beta_2(\widehat{t_4}))}{4\lambda_1}$$
$$\geq \frac{\lambda_2\beta_2(t_4)(1 - \alpha(t_4)\beta_2(t_4))}{500\lambda_1}.$$

Since $\beta_2(0) \geq \alpha(0)$, we can induce that $\beta_2(t) \geq \alpha(t)$ for all $t$. Since $\beta_2$ will increase for all time, we know $\beta_2(t_4) \geq \frac{\sqrt{6\eta\lambda_1}}{20}$.

So we get $\alpha(t_4)\beta_2(t_4) \geq \frac{3}{20}$. Then $t_4$ will satisfy initialization condition. Therefore, we can repeat the above proof from $t_4$. So we know that Proposition E.1 is true for $t \leq t^\dagger + 1$. Then we obtain that $t^\dagger = \infty$ and $\alpha(t)\beta_2(t) < 1$ for all $t$. $\qquad\square$

### E.1.1 Proof of Lemma 4.2

Since we have proved that $\alpha(t)\beta_2(t) < 1$ for all $t$, according to the update rule of $\beta_2$ in (E.4), we can conclude that $\beta_2(t + 1) > \beta_2(t)$ for all $t$. This completes the proof of Lemma 4.2 (iii).

Next based on Lemma 4.2 (iii), we will prove Lemma 4.2 (i). Recall that the Hessian matrix for $\theta = (\alpha, \beta_1, \beta_2)^T$ is given by:

$$H(\theta) = \begin{pmatrix} \lambda_1\beta_1^2 + \lambda_2\beta_2^2 & 2\lambda_1\alpha\beta_1 & 2\lambda_2\alpha\beta_2 - \lambda_2 \\ 2\lambda_1\alpha\beta_1 & \lambda_1\alpha^2 & 0 \\ 2\lambda_2\alpha\beta_2 - \lambda_2 & 0 & \lambda_2\alpha^2 \end{pmatrix}.$$

Since $H(\theta)_{2,2} = \lambda_1\alpha^2$, it is apparent that $S(\theta) \geq \lambda_1\alpha^2$. Let us define matrices $A$ and $B$ as follows:

$$A = \begin{pmatrix} \lambda_1\beta_1^2 & 2\lambda_1\alpha\beta_1 \\ 2\lambda_1\alpha\beta_1 & \lambda_1\alpha^2 \end{pmatrix}, \quad B = \begin{pmatrix} \lambda_2\beta_2^2 & 2\lambda_2\alpha\beta_2 - \lambda_2 \\ 2\lambda_2\alpha\beta_2 - \lambda_2 & \lambda_2\alpha^2 \end{pmatrix}.$$

We use $\|M\|_2$ to denote the maximum singular value of any matrix $M$. It is apparent that:

$$S(\theta) \leq \|A\|_2 + \|B\|_2.$$

Since we have proved that $0 < \alpha\beta_2 < 1$, for $B$ we have:

$$\|B\|_2 = \max_{-1 \leq t \leq 1}\left\{\left|\lambda_2\beta_2^2 t^2 + \lambda_2\alpha_2^2(1-t^2) + 2\lambda_2(2\alpha\beta_2 - 1)t\sqrt{1-t^2}\right|\right\}$$
$$\leq \lambda_2\beta_2^2 + \lambda_2\alpha^2 + \lambda_2.$$

Thus we have:

$$\frac{\|B\|_2}{\lambda_1\alpha^2} \leq \frac{\lambda_2}{\lambda_1}(1 + \frac{1}{\alpha^4}) + \frac{\lambda_2}{\lambda_1\alpha^2} \leq \frac{\lambda_2}{\lambda_1}(1 + \lambda_1^2\eta^2) + \lambda_2\eta$$
$$\leq \frac{\lambda_2}{\lambda_1} + \frac{\sqrt{\lambda_2}}{10\sqrt{\lambda_1}} + \lambda_1\lambda_2\eta^2 \leq 0.0111.$$

It is also easy to see from the above equation that:

$$\mathrm{Tr}(B) = \lambda_2\beta_2^2 + \lambda_2\alpha^2 \leq 0.012 \cdot \lambda_1\alpha^2.$$

For $\|A\|_2$, without loss of generality, suppose $\beta_1 \geq 0$. Since we already know that:

$$\alpha \geq \frac{1}{\sqrt{\lambda_1}\eta} \geq \sqrt{10} \cdot (\frac{\lambda_2}{\lambda_1})^{\frac{1}{4}} > 6\beta_1,$$

let $u = (t, \sqrt{1-t^2})$ with $t \in [-1, 1]$. We then have:

$$|u^T Au| \leq \lambda_1\alpha^2\left[t^2 + \frac{\beta_1^2}{\alpha^2}(1-t^2) + 4\frac{\beta_1}{\alpha}t\sqrt{1-t^2}\right].$$

Thus we have:

$$\frac{|u^T Au|}{\lambda_1\alpha^2} \leq t^2 + \frac{1}{36}(1-t^2) + \frac{2}{3}t\sqrt{1-t^2} \leq 1.105.$$

Therefore, we get $\|A\|_2 \leq 1.105 \cdot \lambda_1\alpha^2$. Consequently, we obtain:

$$S(\theta) \leq \|A\|_2 + \|B\|_2 \leq (0.0111 + 1.105) \cdot \lambda_1\alpha^2 \leq 1.12 \cdot \lambda_1\alpha^2.$$

Thus we have proved Lemma 4.2 (i).

Next we will prove Lemma 4.2 (ii) based on Lemma 4.2 (i)(iii). Following the notation in the previous section, we aim to estimate the second eigenvalue of $A$, denoted as $k_1$. We have:

$$k_1 = \mathrm{tr}(A) - \lambda_{\max}(A).$$

It is straightforward to prove that $\lambda_{\max}(A) \geq \lambda_1\alpha^2 + \lambda_1\beta_1^2 = \mathrm{tr}(A)$; thus, $k_1 \leq 0$. Let $k_2$ denote the second eigenvalue of $B$. We have:

$$k_2 \geq -\left\|\begin{pmatrix} 0 & \lambda_2\alpha\beta_2 - \lambda_2 \\ \lambda_2\alpha\beta_2 - \lambda_2 & 0 \end{pmatrix}\right\|_2 \geq -\lambda_2.$$

Therefore, if $H(\theta)$ has any negative eigenvalue $k$, then $k \geq k_1 - \lambda_2$. Let $v_1$, $v_2$, and $v_3$ denote the three eigenvectors of $H(\theta)$ and $u = (0, 1, 0)^T$. We express $u$ as:

$$u = \phi_1 v_1 + \phi_2 v_2 + \phi_3 v_3,$$

where $\phi_1^2 + \phi_2^2 + \phi_3^2 = 1$. We then have:

$$\lambda_1\alpha^2 = u^T H(\theta)u \leq \phi_1^2\lambda_{\max}(H(\theta)) + (\mathrm{tr}(H(\theta)) - \lambda_{\max}(H(\theta)) - k)(\phi_2^2 + \phi_3^2)$$
$$\leq \phi_1^2(\lambda_{\max}(A) + \lambda_{\max}(B)) + (\mathrm{tr}(H(\theta)) - \lambda_{\max}(A) - k_1 + \lambda_2)(1 - \phi_1^2)$$
$$\leq \phi_1^2(1.105\lambda_1\alpha^2 + 0.01201\lambda_1\alpha^2) + (\mathrm{tr}(B) + \lambda_2)(1 - \phi_1^2)$$
$$\leq \phi_1^2 \cdot 1.12\lambda_1\alpha^2 + (0.012\lambda_1\alpha^2 + 0.001\lambda_1\alpha^2)(1 - \phi_1^2)$$
$$= \lambda_1\alpha^2\left(1.12\phi_1^2 + 0.013(1 - \phi_1^2)\right).$$

Thus $1.12\phi_1^2 + 0.013(1 - \phi_1^2) \geq 1$. This implies $\phi_1 \geq 0.9$, which means $|\cos(v_1, u)| \geq 0.9$. Therefore, we have completed the proof of Lemma 4.2 (ii).

### E.1.2 Proof of Theorem 4.3

Theorem 4.3 is a direct corollary of Proposition E.1 and Lemma 4.2 (ii).

### E.2 Proof of Theorem 4.1

Based on Lemma 4.2, we have $\beta_2 > 0$. Using the condition that $\alpha(t)\beta_2(t) < 1$ and the lower bound of $\alpha$ in Theorem 4.3, we can conclude that for any $\epsilon > 0$, there exists $t_\epsilon$ such that:

$$\alpha(t)\beta_2(t) \geq 1 - \epsilon \quad \text{for} \quad t \geq t_\epsilon. \tag{E.8}$$

Otherwise, $\alpha(t)\beta_2(t)$ would approach infinity.

If there exists $T^\star$ such that:

$$\lambda_1 \eta \alpha^2 \geq 2 \quad \text{for} \quad t \geq T^\star, \tag{E.9}$$

then we have:

$$\beta_1^2(t) \geq \beta_1^2(T^\star) > 0 \quad \text{for} \quad t \geq T^\star.$$

Based on $\alpha \geq \sqrt{\frac{2}{\lambda_1 \eta}}$ and $\alpha(t)\beta_2(t) < 1$, we can derive $\beta_2(t) < \sqrt{\frac{\lambda_1 \eta}{2}}$. Using equation (E.8), it is straightforward to prove that there exists $T_1 > T^\star$ such that:

$$\lambda_2 \beta_2(t)(1 - \alpha(t)\beta_2(t)) \leq \frac{\lambda_1 \beta_1^2(T^\star)\alpha(t)}{2} \quad \text{for} \quad t \geq T_1.$$

Thus, we have:

$$\alpha(t+1) \leq \alpha(t) - \frac{\lambda_1 \beta_1^2(T^\star)\alpha(t)}{2},$$

which contradicts equation (E.9). Therefore, we have proven that there exist infinitely many $t$ such that $\lambda_1 \eta \alpha^2 < 2$.

If there exists a $\delta > 0$ which does not satisfy the above proposition, then for any $t > t_0$, we have $\beta_2(t) < \sqrt{\frac{\lambda_1 \eta}{2+\delta}}$. Combining this with our previous conclusion, we find that there exist infinitely many $t$ such that:

$$\alpha(t)\beta_2(t) < \sqrt{\frac{2}{\lambda_1 \eta}} \cdot \sqrt{\frac{\lambda_1 \eta}{2+\delta}} = \sqrt{\frac{2}{2+\delta}}.$$

This immediately contradicts Equation (E.8) when we set $\epsilon = 1 - \sqrt{\frac{2}{2+\delta}}$. So we have now proved that for any $\delta > 0$ and $\epsilon > 0$, there exists a time $\widehat{T}(\delta, \epsilon)$ such that for any $t \geq \widehat{T}(\delta, \epsilon)$:

$$\lambda_1 \alpha^2(t) \leq 2 + \delta, \quad 1 - \alpha(t)\beta_2(t) \leq \epsilon. \tag{E.10}$$

Finally, we will prove that for any $d > 0$, there exists a time $T_d$ such that $\beta_1^2(t) < d$ for $t > T_d$. If this were not the case, there would be infinitely many $t$ such that $\lambda_1 \eta \alpha^2(t) \geq 1.9$. This implies that $\beta_2(t) < \sqrt{\frac{\lambda_1 \eta}{1.9}}$ for any $t$. We set $\epsilon_1, \delta > 0$ such that:

$$\left(1 + \frac{\lambda_2 \epsilon_1(2+\delta)}{\lambda_1(1-\epsilon_1)}\right)\left(1 - \lambda_1 \eta d + \frac{\lambda_1 \eta^2 \epsilon_1}{\sqrt{1.9}}\right) = y < 1.$$

When $t > \widehat{T}(\delta, \epsilon)$ and $\beta_2^2(t) \geq d$, we have:

$$\alpha(t+1)\beta_2(t+1) \leq \left(1 + \frac{\lambda_2 \epsilon_1(2+\delta)}{\lambda_1(1-\epsilon_1)}\right)\left(1 - \lambda_1 \eta d + \frac{\lambda_1 \eta^2 \epsilon_1}{\sqrt{1.9}}\right) = y.$$

This immediately contradicts our previous conclusion when we set $\epsilon = 1 - y > 0$. Therefore, for any $d > 0$, there exists a time $T_d$ such that $\beta_2^2(t) < d$ for $t > T_d$. Combining this with Equation (E.10), we can conclude that for any $\epsilon \geq 0$, there exists a time $T_\epsilon$ such that for $t \geq T_\epsilon$, $L(\theta(t)) \leq \epsilon$.

We now know that for any $\epsilon$ and $\delta$, there exists $\widetilde{T}(\delta, \epsilon)$ such that for $t \geq \widetilde{T}(\delta, \epsilon)$, we have:

$$\lambda_1 \eta \alpha^2(t) \leq 2 + \delta, \quad |\beta_1(t)| \leq \epsilon.$$

Recall that for $\theta = (\alpha, \beta_1, \beta_2)^T$, we have:

$$S(\theta) = \max \lambda_1 (t_2 \alpha + t_1 \beta_1)^2 + \lambda_2 (t_3 \alpha + t_1 \beta_2)^2 \quad \text{subject to} \quad t_1^2 + t_2^2 + t_3^2 \leq 1.$$

When $\beta_1 = 0$, $\alpha > 0$, $1 \leq \lambda_1 \eta \alpha^2 \leq 2 + \delta$, and $0 \leq \beta_2 \leq \frac{1}{\alpha}$, we have:

$$S(\theta) = \max \lambda_1 \alpha^2 t_2^2 + \lambda_2 (t_3 \alpha + t_1 \beta_2)^2 \quad \text{subject to} \quad t_1^2 + t_2^2 + t_3^2 \leq 1$$
$$\leq \max \lambda_1 \alpha^2 t_2^2 + \lambda_2 (\alpha^2 + \beta_2^2)(t_1^2 + t_3^2) \quad \text{subject to} \quad t_1^2 + t_2^2 + t_3^2 \leq 1.$$

Since $\lambda_2 \beta_2^2 \leq \frac{\lambda_2}{\alpha^2} \leq \lambda_1 \lambda_2 \eta \leq \frac{1}{100\eta} \leq \frac{\lambda_1 \alpha^2}{100}$, we get that $\lambda_2 \alpha^2 + \lambda_2 \beta_2^2 < \lambda_1 \alpha^2$. Thus, when $\beta_1 = 0$, $\alpha > 0$, $1 \leq \lambda_1 \eta \alpha^2 \leq 2 + \delta$, and $0 \leq \beta_2 \leq \frac{1}{\alpha}$, we have $\|H(\theta)\|_2 = \lambda_1 \alpha^2 \leq \frac{2+\delta}{\eta}$.

We can see that set $A = \{(\alpha, \beta_2, t_1, t_2, t_3) \,|\, \alpha > 0, 1 \leq \lambda_1 \eta \alpha^2 \leq 2 + \delta, 0 \leq \beta_2 \leq \frac{1}{\alpha}, t_1^2 + t_2^2 + t_3^2 \leq 1\}$ is compact in $\mathbb{R}^5$. Now we define:

$$F(\alpha, \beta_1, \beta_2, t_1, t_2, t_3) = \lambda_1 (t_2 \alpha + t_1 \beta_1)^2 + \lambda_2 (t_3 \alpha + t_1 \beta_2)^2,$$
$$g(\beta_1) = \max_{(\alpha, \beta_2, t_1, t_2, t_3) \in A} F(\alpha, \beta_1, \beta_2, t_1, t_2, t_3).$$

It is apparent that $F$ is a continuous function. Thus, based on Berge's Maximum Theorem, $g$ is also a continuous function. Since $g(0) \leq 2 + \delta$, we know there exists $\epsilon_\delta$ such that when $|\beta_1(t)| \leq \epsilon_\delta$, we have $g(\beta_1) \leq (2 + 2\delta)/\eta$. Therefore, we know that for $t \geq \widetilde{T}(\delta, \epsilon_\delta)$, $\|H(\theta(t))\|_2 \leq (2 + 2\delta)/\eta$, thus completing the proof.

### E.3 Proof of Theorem 4.4

We consider the following initialization:

$$1.1 \leq \eta \lambda_1 \alpha^2(0) \leq 1.5, \quad \beta_2(0) \geq \max \left\{ \frac{\sqrt{6\eta\lambda_1}}{20}, \frac{3}{20\alpha(0)}, \alpha(0) \right\},$$

$$\frac{\lambda_2 \beta_2(0)}{500 \lambda_1 \alpha(0)} (1 - \alpha(0)\beta_2(0)) \leq \beta_1^2(0) \leq \frac{\lambda_2 \beta_2(0)}{50 \lambda_1 \alpha(0)} (1 - \alpha(0)\beta_2(0)).$$

Based on the result and proof in Theorem 4.3, we know that for any $t \geq 0$, $\alpha(t)\beta_2(t) \geq 0.15$.

Let $T$ be the largest time such that $\alpha$ consistently increases before $T$, $T_1$ be the largest time such that $\alpha(t)\beta_2(t) \leq 0.85$ for $t \leq T_1$, $T_2$ be the first time such that $\lambda_1 \eta \alpha^2 \geq 2$, and $T_3$ be the first time such that

$$\beta_1^2(t) > \frac{\lambda_2 \beta_2(t)}{50 \lambda_1 \alpha(t)} (1 - \alpha(t)\beta_2(t)).$$

Since $\beta_1^2(1) \leq \frac{1}{4} \beta_1^2(0)$, we know $T_3 > \min\{T_1, T_2\}$. For $t \leq T_3$, we have the following equation:

$$\frac{\beta_2(t)}{\alpha(t)} \leq \max \left\{ \frac{\beta_2(0)}{\alpha(0)}, \frac{50}{49} \right\}.$$

Thus we have:

$$\alpha(t)\beta_2(t) \leq \frac{50}{49} \cdot \frac{\alpha^2(t)}{\alpha^2(0)} \cdot \alpha(0)\beta_2(0).$$

Then, based on the range of $\alpha(0)$ and the range of $\alpha^2(t)$ in the previous theorem, we have $\alpha(T_3)\beta_2(T_3) \leq 0.78$, which yields $T_1 > T_3$. Thereby, we get $T_2 < T_3 < T_1$. We also know $T > T_3$. Based on the definition of $T_1$, we know:

$$\frac{\max_{0 \leq t \leq T_2} \alpha(t)\beta_2(t) (1 - \alpha(t)\beta_2(t))}{\min_{T_2 \leq t \leq T_3} \alpha(t)\beta_2(t)(1 - \alpha(t)\beta_2(t))} \leq 2.$$

Then we can easily obtain:

$$\frac{\max_{0 \le t \le T_2}(\alpha^2(t+1) - \alpha^2(t))}{\min_{T_2 < t \le T_3}(\alpha^2(t+1) - \alpha^2(t))} \le 2.1.$$

Similar to the previous proof, we set $n_3$ and $n_4$ such that:

$$100\eta(\alpha^2(t+1) - \alpha^2(t)) \le \frac{1}{n_3} \quad \text{for} \quad 0 \le t \le T_2,$$

$$100\eta(\alpha^2(t+1) - \alpha^2(t)) \ge \frac{1}{n_4} \quad \text{for} \quad T_2 < t \le T_3.$$

Then we have $\min\{n_3, n_4\} \ge 100$ and $\frac{n_3}{n_4} \ge \frac{1}{2.1}$.

Similar to the previous proof, we obtain:

$$\left| \prod_{i=0}^{T_3} (100\eta\alpha^2(i) - 1) \right| \le \left( \frac{2^{\frac{1}{2}}}{e^{\frac{1}{2}}} \right)^{n_3} \cdot \left( \frac{\lambda^\lambda}{e^{\lambda-1}} \right)^{n_4} \cdot \sqrt{2\lambda} \cdot e^{\frac{10n_4 + n_3}{12 n_3 n_4}} \le 0.1 \quad \text{when} \quad \lambda = 1.37.$$

If $\lambda_1 \eta \alpha^2(T_3) \le 2.37$, based on the fact that $T_3 \le T_1$, we have:

$$\begin{aligned}
\beta_1^2(T_3) &\le \frac{1}{100} \cdot \frac{\lambda_2 \beta_2(0)}{50\lambda_1 \alpha(0)}(1 - \alpha(0)\beta_2(0)) = \frac{1}{100} \cdot \frac{\lambda_2 \alpha(0)\beta_2(0)(1 - \alpha(0)\beta_2(0))}{50\lambda_1 \alpha^2(0)} \\
&\le \frac{1}{40} \cdot \frac{\lambda_2 \alpha(0)\beta_2(0)(1 - \alpha(0)\beta_2(0))}{50\lambda_1 \alpha^2(T_3)} \le \frac{1}{20} \cdot \frac{\lambda_2 \alpha(T_3)\beta_2(T_3)(1 - \alpha(T_3)\beta_2(T_3))}{50\lambda_1 \alpha^2(T_3)} \\
&= \frac{1}{20} \cdot \frac{\lambda_2 \beta_2(T_3)(1 - \alpha(T_3)\beta_2(T_3))}{50\lambda_1 \alpha(T_3)}.
\end{aligned}$$

This contradicts the definition of $T_3$. Thus, we conclude that $\lambda_1 \eta \alpha^2(T_3) > 2.37$, which implies $\lambda_1 \eta \alpha^2(T) > 2.37$ and $S(\theta(T)) > \frac{2.37}{\eta}$.

Since $\lambda_1 \eta \alpha^2(t) > 2.37$ for $T_3 \le t \le T$, if $T \ge T_3 + 9$, we have either $\beta_1^2(T_3 + 8) \ge c^2 = \frac{\sqrt{\lambda_2}}{4\sqrt{\lambda_1}}$ or:

$$\begin{aligned}
\beta_1^2(T_3 + 8) &\ge 1.37^{16} \cdot \frac{\lambda_2 \beta_2(T_3)(1 - \alpha(T_3)\beta_2(T_3))}{50\lambda_1 \alpha(T_3)} \ge \frac{3\lambda_2 \beta_2(T_3)(1 - \alpha(T_3)\beta_2(T_3))}{\lambda_1 \alpha(T_3)} \\
&= \frac{3\lambda_2 \alpha(T_3)\beta_2(T_3)(1 - \alpha(T_3)\beta_2(T_3))}{\lambda_1 \alpha^2(T_3)} \ge \frac{3\lambda_2 \alpha(T_3)\beta_2(T_3)(1 - \alpha(T_3)\beta_2(T_3))}{\lambda_1 \alpha^2(T_3 + 8)} \\
&\ge \frac{3}{2} \cdot \frac{\lambda_2 \alpha(T_3 + 8)\beta_2(T_3 + 8)(1 - \alpha(T_3 + 8)\beta_2(T_3 + 8))}{\lambda_1 \alpha(T_3 + 8)}.
\end{aligned}$$

In the first case, it is also easy to prove that:

$$\lambda_1 \alpha^2(T_3 + 8)\beta_1^2(T_3 + 8) \ge \frac{\sqrt{\lambda_2}}{4\eta\sqrt{\lambda_1}} \ge \frac{\lambda_2}{4} \ge \lambda_2 \alpha(T_3 + 8)\beta_2(T_3 + 8)(1 - \alpha(T_3 + 8)\beta_2(T_3 + 8)).$$

This shows that $T \le T_3 + 8$, which contradicts our assumption. Therefore, $T \le T_3 + 8$. Based on the limited movement speed of $\alpha\beta_2$, we can easily prove that $\alpha(T)\beta_2(T) \le \alpha(T_3)\beta_2(T_3) + 4\eta^2 \le 0.82$. So we know that $T_1 \ge T$.

Let $T_4 \ge T$ be the first time such that

$$\beta_1^2(T_4) > \eta\lambda_2 \alpha(T)\beta_2(T)(1 - \alpha(T)\beta_2(T)).$$

Note that for the constraint constant $c$, we have:

$$c^2 = \frac{\sqrt{\lambda_2}}{4\sqrt{\lambda_1}} > \frac{\eta\lambda_2}{4} > \eta\lambda_2 \alpha(T)\beta_2(T)(1 - \alpha(T)\beta_2(T)).$$

Let $T_5$ be the first time after $T$ such that $\lambda_1 \eta \alpha^2 < 2$. Note that $T_5 < \infty$; otherwise, $L(\theta)$ cannot converge, which contradicts Theorem 4.1. For $T \le t < T_4$, we have:

$$\alpha(t+1) \le \alpha(t) + \eta\lambda_2 \beta_2(t)(1 - \alpha(t)\beta_2(t)),$$

$$\alpha(t+1) \ge \alpha(t) - \frac{1}{4}\eta^2 \lambda_1 \lambda_2 \alpha(t) \ge \alpha(t) - \frac{1}{400}\alpha(t).$$

Thus, for $T \le t < T_4$, we have:

$$\lambda_1 \eta \alpha^2 (t+1) \ge (1 - \frac{1}{400})^2 \lambda_1 \eta \alpha^2 (t). \qquad \text{(E.11)}$$

We already know that

$$\beta_1^2(T) > \frac{\lambda_2 \beta_2(T)}{\lambda_1 \alpha(T)}(1 - \alpha(T)\beta_2(T)).$$

If $T_4 \ge T + 3$, let $z = \lambda_1 \eta \alpha^2(T) \ge 2.37$. Then we have the following property:

$$(z-1)^2((1 - \frac{1}{400})^2 z - 1)^2 \ge z.$$

Using this property, we will have either $\beta_1^2(T+2) \ge \frac{\sqrt{\lambda_2}}{4\sqrt{\lambda_1}}$ or:

$$\beta_1^2(T+2) \ge (z-1)^2 \left((1 - \frac{1}{400})^2 z - 1\right)^2 \beta_1^2(T) \ge \lambda_1 \eta \alpha^2(T) \beta_1^2(T)$$
$$> \lambda_2 \eta \alpha(T) \beta_2(T)(1 - \alpha(T)\beta_2(T)).$$

Thus, we get $T_4 \ge T+2$, which leads to a contradiction. Therefore, $T_4 \le T+2$. Then, using (E.11), we know $T_5 > T_4$.

For $t \le T_4 < T_5$, we have

$$1 - \frac{1}{400} \le \frac{\alpha(t+1)}{\alpha(t)} = 1 + \frac{\eta \lambda_2 \beta_2(t)(1 - \alpha(t)\beta_2(t))}{\alpha(t)} \le 1 + \frac{\eta^2 \lambda_1 \lambda_2 \alpha(t)\beta_2(t)(1 - \alpha(t)\beta_2(t))}{\lambda_1 \eta \alpha^2(t)} \le 1 + \frac{1}{800}.$$

$$1 \le \frac{\beta_2(t+1)}{\beta_2(t)} \le 1 + \eta \lambda_2.$$

Based on the fact that $0.15 \le \alpha(T)\beta_2(T) \le 0.82$ and $\beta_1^2(T+1) \ge 1.37^2 \beta_1^2(T)$, it is easy to prove that for $T \le t < T_4$, we have $\alpha(t+1) < \alpha(t)$.

For $T_4 \le t < T_5$, we have:

$$\beta_1^2(t) \ge \beta_1^2(T_4) > \lambda_2 \eta \alpha(T) \beta_2(T)(1 - \alpha(T)\beta_2(T)).$$

Since $0.15 \le \alpha(T)\beta_2(T) \le 0.85$, we know:

$$\frac{\lambda_2 \beta_2(t)}{\lambda_1 \alpha(t)}(1 - \alpha(t)\beta_2(t)) = \frac{\eta \lambda_2 \alpha(t)\beta_2(t)(1 - \alpha(t)\beta_2(t))}{\lambda_1 \eta \alpha^2(t)}$$

$$\le \frac{\alpha(t)\beta_2(t)(1 - \alpha(t)\beta_2(t))}{\alpha(T)\beta_2(T)(1 - \alpha(T)\beta_2(T))} \cdot \frac{\lambda_1 \eta \alpha^2(T)}{\lambda_1 \eta \alpha^2(t)} \cdot \frac{\eta \lambda_2 \alpha(T)\beta_2(T)(1 - \alpha(T)\beta_2(T))}{\lambda_1 \eta \alpha^2(T)}$$

$$\le 2 \cdot \frac{\lambda_1 \eta \alpha^2(T)}{2} \cdot \frac{\eta \lambda_2 \alpha(T)\beta_2(T)(1 - \alpha(T)\beta_2(T))}{\lambda_1 \eta \alpha^2(T)}$$

$$= \lambda_2 \eta \alpha(T)\beta_2(T)(1 - \alpha(T)\beta_2(T)) < \beta_1^2(t).$$

Thus, for $T_4 \le t < T_5$, we have $\alpha(t+1) < \alpha(t)$. Therefore, we conclude that for $T \le t < T_5$, we have $\alpha(t+1) < \alpha(t)$.

Let $T_5 < T_6 \le \infty$ be the first time after $T_5$ such that $\alpha(t+1) \ge \alpha(t)$. We know that $\lambda_1 \alpha^2(T_6) < \frac{2}{\eta}$. Recall that:

$$H(\theta) = \begin{pmatrix} \lambda_1 \beta_1^2 + \lambda_2 \beta_2^2 & 2\lambda_1 \alpha \beta_1 & 2\lambda_2 \alpha \beta_2 - \lambda_2 \\ 2\lambda_1 \alpha \beta_1 & \lambda_1 \alpha^2 & 0 \\ 2\lambda_2 \alpha \beta_2 - \lambda_2 & 0 & \lambda_2 \alpha^2 \end{pmatrix}.$$

If $T_6 = \infty$, then we know that $\|H(\theta(T_6))\|_2 = \lambda_1 \alpha^2(T_6) < \frac{2}{\eta}$.

For the following part, we suppose $T_6 < \infty$. At $T_6$, we have:

$$\lambda_1 \alpha(T_6) \beta_1^2(T_6) \le \lambda_2 \beta(T_6)(1 - \alpha(T_6)\beta_2(T_6)).$$

If $\lambda_1 \alpha^2(T_6) \leq \frac{1.75}{\eta}$, then we directly have $\|H(\theta(T_6))\| \leq 1.12 \cdot \frac{1.75}{\eta} < \frac{2}{\eta}$. Next we suppose $\lambda_1 \alpha^2(T_6) \geq \frac{1.75}{\eta}$. We then have:

$$\beta_1^2(T_6) \leq \frac{\lambda_2 \alpha(T_6)\beta_2(T_6)(1 - \alpha(T_6)\beta_2(T_6))}{\lambda_1 \alpha^2(T_6)} \leq \frac{\lambda_2 \eta}{7}.$$

In the following proof, for conciseness, we use $\theta$ to denote $\theta(T_6)$. Let $u = (t_1, t_2, t_3)^T$. We have:

$$u^T H(\theta) u \leq (\lambda_1 \beta_1^2 + \lambda_2 \beta_2^2) t_1^2 + \lambda_1 \alpha^2 t_2^2 + \lambda_1 \alpha^2 t_3^2 + 4\lambda_1 \alpha \beta_1 t_1 t_2 + 2\lambda_2(2\alpha\beta_2 - 1)t_1 t_3$$

$$\leq (\lambda_1 \beta_1^2 + \lambda_2 \beta_2^2 + \lambda_2) t_1^2 + \lambda_1 \alpha^2 t_2^2 + (\lambda_2 \alpha^2 + \lambda_2) t_3^2 + 4\lambda_1 \alpha \beta_1 t_1 t_2$$

$$\leq (\lambda_1 \beta_1^2 + \lambda_2 \beta_2^2 + \lambda_2) t_1^2 + \lambda_1 \alpha^2 t_2^2 + (\lambda_2 \alpha^2 + \lambda_2) t_3^2 + 5\lambda_1 \beta_1^2 t_2^2 + \frac{4\lambda_1 \alpha^2 t_1^2}{5}$$

$$= (\lambda_1 \beta_1^2 + \lambda_2 \beta_2^2 + \lambda_2 + \frac{4}{5} \cdot \lambda_1 \alpha^2) t_1^2 + (\lambda_1 \alpha^2 + 5\lambda_1 \beta_1^2) t_2^2 + (\lambda_2 \alpha^2 + \lambda_2) t_3^2.$$

We note that:

$$\lambda_2 \alpha^2 + \lambda_2 < \frac{2}{\eta},$$

$$\lambda_1 \beta_1^2 + \lambda_2 \beta_2^2 + \lambda_2 + \frac{4}{5} \cdot \lambda_1 \alpha^2 \leq \frac{\eta}{7} + \frac{\lambda_2}{\alpha^2} + \lambda_2 + \frac{4}{5} \cdot \lambda_1 \alpha^2 < \frac{2}{\eta},$$

$$\lambda_1 \alpha^2 + 5\lambda_1 \beta_1^2 \leq \frac{2}{\eta} + \frac{5\eta \lambda_1 \lambda_2}{7} < \frac{2.01}{\eta}.$$

Thus, we know that for any $u \in \mathbb{R}^3$, $u^T H(\theta) u \leq \frac{2.01}{\eta}$. Therefore, $S(\theta(T_6)) \leq \frac{2.01}{\eta}$.

## E.4 Proof for Lemma 4.5

We have the following inequalities:

$$\frac{1 - \sqrt{\frac{4.2}{\lambda_1 \eta}}\beta_2(t)}{1 - \sqrt{\frac{2}{\lambda_1 \eta}}\beta_2(t)} \leq \frac{1 - \alpha(t)\beta_2(t)}{1 - \sqrt{\frac{2}{\lambda_1 \eta}}\beta_2(t)} \leq \frac{1 - \sqrt{\frac{1.1}{\lambda_1 \eta}}\beta_2(t)}{1 - \sqrt{\frac{2}{\lambda_1 \eta}}\beta_2(t)}.$$

Using the condition $\sqrt{\frac{2}{\lambda_1 \eta}}\beta_2(t) \leq \frac{1}{2}$, we obtain:

$$\frac{1 - \sqrt{\frac{4.2}{\lambda_1 \eta}}\beta_2(t)}{1 - \sqrt{\frac{2}{\lambda_1 \eta}}\beta_2(t)} \geq 2\left(1 - \frac{1}{2}\sqrt{\frac{4.2}{2}}\right),$$

$$\frac{1 - \sqrt{\frac{1.1}{\lambda_1 \eta}}\beta_2(t)}{1 - \sqrt{\frac{2}{\lambda_1 \eta}}\beta_2(t)} \leq 2\left(1 - \frac{1}{2}\sqrt{\frac{1.1}{2}}\right).$$

Therefore, we can conclude:

$$0.75 \leq \frac{1}{4\left(1 - \frac{1}{2}\sqrt{\frac{1.1}{2}}\right)^2} \leq \frac{\widehat{L}(\theta)}{L_2(\theta)} \leq \frac{1}{4\left(1 - \frac{1}{2}\sqrt{\frac{4.2}{2}}\right)^2} \leq 3.3.$$

Moreover, when $\alpha(t) \leq \sqrt{\frac{2}{\lambda_1 \eta}}$, we have $\widehat{L}(\theta) \leq L_2(\theta)$.

## E.5 Proof of Theorem 4.6

Based on the initialization and Theorem 4.3, we know that $\frac{\sqrt{2}\beta_2(t)}{\sqrt{\lambda_1 \eta}} \geq 0.15$ for $t \geq 0$. According to the condition given by Theorem 4.3, we only need to consider $t$ such that $\frac{\sqrt{2}\beta_2(t)}{\sqrt{\lambda_1 \eta}} \leq 0.5$. Using this condition, we can derive:

$$\frac{1}{2} \cdot \frac{\sqrt{2}}{\sqrt{\lambda_1 \eta}}\left(1 - \frac{\sqrt{2}\beta_2(t)}{\sqrt{\lambda_1 \eta}}\right) \leq \alpha(t)(1 - \alpha(t)\beta_2(t)) \leq 2 \cdot \frac{\sqrt{2}}{\sqrt{\lambda_1 \eta}}\left(1 - \frac{\sqrt{2}\beta_2(t)}{\sqrt{\lambda_1 \eta}}\right).$$

Thus we have:

$$\beta_2(t) + \lambda_2 \frac{\sqrt{\eta}}{\sqrt{2\lambda_1}} \left(1 - \frac{\sqrt{2}\beta_2(t)}{\sqrt{\lambda_1\eta}}\right) \leq \beta_2(t+1) \leq \beta_2(t) + \lambda_2 \frac{2\sqrt{2\eta}}{\sqrt{\lambda_1}} \left(1 - \frac{\sqrt{2}\beta_2(t)}{\sqrt{\lambda_1\eta}}\right).$$

Then we obtain:

$$\left(1 - \frac{4\lambda_2}{\lambda_1}\right)\left(1 - \frac{\sqrt{2}\beta_2(t)}{\sqrt{\lambda_1\eta}}\right) \leq 1 - \frac{\sqrt{2}\beta_2(t+1)}{\sqrt{\lambda_1\eta}} \leq \left(1 - \frac{\lambda_2}{\lambda_1}\right)\left(1 - \frac{\sqrt{2}\beta_2(t)}{\sqrt{\lambda_1\eta}}\right).$$

Finally, we have:

$$\left(1 - \frac{4\lambda_2}{\lambda_1}\right)^2\left(1 - \frac{\sqrt{2}\beta_2(t)}{\sqrt{\lambda_1\eta}}\right)^2 \leq \left(1 - \frac{\sqrt{2}\beta_2(t+1)}{\sqrt{\lambda_1\eta}}\right)^2 \leq \left(1 - \frac{\lambda_2}{\lambda_1}\right)^2\left(1 - \frac{\sqrt{2}\beta_2(t)}{\sqrt{\lambda_1\eta}}\right)^2.$$

Therefore, we can observe that the decrease speed is $\exp\{-\frac{k\lambda_2}{\lambda_1}\}$ for $1 \leq k \leq 4$, and it is independent of the learning rate $\eta$. This indicates that the condition number of the input covariance matrix significantly affects the training speed in EoS regime.

## F Proof of Section 5

### F.1 Proof of Lemma 5.1

First, we consider a minimal $\theta^\star = (\alpha, \beta_1, \beta_2)$. We have $\beta_1 = 0$ and $\alpha\beta_2 = 1$. Let $u = (t_1, t_2, t_3) \in \mathbb{R}^3$ such that $\|u\|_2 = 1$. Then we have:

$$u^T H(\theta^\star)u = \lambda_1(t_2\alpha)^2 + \lambda_2(t_3\alpha + t_1\beta_2)^2 \leq \lambda_1\alpha^2 t_2^2 + \lambda_2(\alpha^2 + \beta_2^2)(t_1^2 + t_3^2).$$

Since $t_1^2 + t_2^2 + t_3^2 = 1$, we can see:

$$u^T H(\theta^\star)u \leq \lambda_1\alpha^2 t_2^2 + \lambda_2(\alpha^2 + \beta_2^2)(1 - t_2^2) \leq \max\{\lambda_1\alpha^2, \lambda_2\alpha^2 + \lambda_2\beta_2^2\}.$$

Other other hand, when $u = (0, 1, 0)$ we have $S(\theta) = \lambda_1\alpha^2$. When $t_1 = \frac{\beta_2}{\sqrt{\alpha^2+\beta_2^2}}, t_3 = \frac{\alpha}{\sqrt{\alpha^2+\beta_2^2}}$, we have $S(\theta) = \lambda_2\alpha^2 + \lambda_2\beta_2^2$. Therefore, we know that

$$S(\theta) = \max\{\lambda_1\alpha^2, \lambda_2\alpha^2 + \lambda_2\beta_2^2\}.$$

Thus, we have

$$S(\theta) = \lambda_1\alpha^2 \Leftrightarrow \lambda_1\alpha^2 \geq \lambda_2\alpha^2 + \lambda_2\beta_2^2 \Leftrightarrow \alpha^4 \geq \frac{\lambda_2}{\lambda_1 - \lambda_2}$$

$$\Leftrightarrow \alpha^2 - \beta_2^2 \geq \sqrt{\frac{\lambda_2}{\lambda_1 - \lambda_2}} - \sqrt{\frac{\lambda_1 - \lambda_2}{\lambda_2}}.$$

Now we consider gradient flow (GF) and let $\phi(\theta(0)) = (\alpha(\infty), \beta_1(\infty), \beta_2(\infty))^T$. Since we know that GF preserves layer norm difference as shown in Theorem 2.1 in Du et al. [2018], we can get:

$$\alpha^2(\infty) - \beta_2^2(\infty) = \alpha^2(0) - \beta_2^2(0) - \beta_1^2(0) \leq 0, \tag{F.1}$$
$$\alpha(\infty)\beta_2(\infty) = 1. \tag{F.2}$$

Note that $\beta_1^2(0) \leq \frac{\lambda_2\eta}{4}$. We can then obtain:

$$\alpha^2(0) - \beta_2^2(0) - \frac{\lambda_2\eta}{4} \leq \alpha^2(\infty) - \beta_2^2(\infty) \leq \alpha^2(0) - \beta_2^2(0). \tag{F.3}$$

Based on the fact that $\alpha^2(0) \geq \frac{1}{\lambda_1\eta}$ and $\alpha(0)\beta_2(0) \leq 1$, we have:

$$\alpha^2(0) - \beta_2^2(0) - \frac{\lambda_2\eta}{4} \geq \frac{10\sqrt{\lambda_2}}{\sqrt{\lambda_1}} - \frac{\sqrt{\lambda_1}}{10\sqrt{\lambda_2}} - \frac{\sqrt{\lambda_2}}{40\sqrt{\lambda_1}} > \sqrt{\frac{\lambda_2}{\lambda_1 - \lambda_2}} - \sqrt{\frac{\lambda_1 - \lambda_2}{\lambda_2}}.$$

Thus we get that $\phi(\theta(0)) = \lambda_1\alpha^2(\infty)$. Now we need to estimate the scale of $\alpha^2(\infty)$.

Recall that $\gamma = \alpha^2(0) - \beta_2^2(0) \le 0$. Using (F.2) and $\alpha^2(\infty) - \beta_2^2(\infty) \le \gamma$, we can get $\alpha^2 \le \frac{\gamma + \sqrt{4+\gamma^2}}{2}$. Using (F.2) and $\alpha^2(\infty) - \beta_2^2(\infty) \ge \gamma - \frac{\sqrt{\lambda_2}}{40\sqrt{\lambda_1}}$, we can get

$$\alpha^2 \ge \frac{\gamma - \frac{\sqrt{\lambda_2}}{40\sqrt{\lambda_1}} + \sqrt{4 + (\gamma - \frac{\sqrt{\lambda_2}}{40\sqrt{\lambda_1}})^2}}{2} \ge \frac{\gamma - \frac{\sqrt{\lambda_2}}{40\sqrt{\lambda_1}} + \sqrt{4 + \gamma^2}}{2}.$$

Thus we can obtain:

$$\frac{\lambda_1(\gamma + \sqrt{4+\gamma^2} - \frac{\sqrt{\lambda_2}}{40\sqrt{\lambda_1}})}{2} \le \phi(\theta(0)) = \lambda_1 \alpha^2(\infty) \le \frac{\lambda_1(\gamma + \sqrt{4+\gamma^2})}{2}.$$

Thus when $\gamma = 0$, we have:

$$\phi(\theta(0)) \ge \lambda_1 - \frac{\sqrt{\lambda_1 \lambda_2}}{40} \ge (1 - \frac{1}{4000})\lambda_1$$

## F.2   Proof of Lemma 5.2

Let $\phi(\theta(t)) = (\alpha(\infty), \beta_1(\infty), \beta_2(\infty))^T$. Using the constraint constant $c = \frac{\lambda_2^{\frac{1}{4}}}{2\lambda_1^{\frac{1}{4}}}$, we obtain:

$$\alpha^2(t) - \beta_2^2(t) - \frac{\sqrt{\lambda_2}}{4\sqrt{\lambda_1}} \le \alpha^2(\infty) - \beta_2^2(\infty) \le \alpha^2(0) - \beta_1^2(0).$$

Similar to the proof of Lemma 5.1, we can derive $\phi(\theta(t)) = \lambda_1 \alpha^2(\infty)$. Using Theorem 4.3, we obtain:

$$\frac{1}{\eta \lambda_1} - \beta_2^2(t) \le \alpha^2(\infty) - \beta_2^2(\infty) \le \frac{4.2}{\eta \lambda_1} - \beta_1^2(t).$$

Following the proof in Lemma 5.1, we can derive:

$$\alpha^2(\infty) \ge \frac{1}{2}\left(\frac{1}{\eta \lambda_1} - \beta_2^2(t) + \sqrt{4 + \left(\frac{1}{\eta \lambda_1} - \beta_2^2(t)\right)^2}\right),$$

$$\alpha^2(\infty) \le \frac{1}{2}\left(\frac{4.2}{\eta \lambda_1} - \beta_2^2(t) + \sqrt{4 + \left(\frac{4.2}{\eta \lambda_1} - \beta_2^2(t)\right)^2}\right).$$

Consequently we obtain:

$$\phi(\theta(t)) \ge \frac{1 - \lambda_1 \eta \beta_2^2(t)}{2\eta} + \frac{\lambda_1 \sqrt{4 + \left(\frac{1}{\eta \lambda_1} - \beta_2^2(t)\right)^2}}{2},$$

$$\phi(\theta(t)) \le \frac{4.2 - \lambda_1 \eta \beta_2^2(t)}{2\eta} + \frac{\lambda_1 \sqrt{4 + \left(\frac{4.2}{\eta \lambda_1} - \beta_2^2(t)\right)^2}}{2}.$$

Since $\beta_2(t)$ is monotonically increasing, both $\frac{1}{\eta \lambda_1} - \beta_2^2(t)$ and $\frac{4.2}{\eta \lambda_1} - \beta_2^2(t)$ decrease monotonically. Given that $\gamma + \sqrt{4+\gamma^2}$ is a monotonically increasing function, we conclude that the two-sided bound of $\phi(\theta(t))$ is a monotonically decreasing function of $t$.

## F.3   Proof of Lemma 5.3

Recall that the definition of $\mathbb{D}$ is:

$$\mathbb{D} = \{\theta : 1 \le \lambda_1 \eta \alpha^2 \le 2; \beta_1 = 0; 0 < \alpha \beta_2 \le 1\}.$$

Using the expression of $H(\theta)$, for $\theta \in \mathbb{D}$, we know $\beta_1 = 0$, and we can obtain:

$$H(\theta) = \begin{pmatrix} \lambda_2 \beta_2^2 & 0 & 2\lambda_2 \alpha \beta_2 - \lambda_2 \\ 0 & \lambda_1 \alpha^2 & 0 \\ 2\lambda_2 \alpha \beta_2 - \lambda_2 & 0 & \lambda_2 \alpha^2 \end{pmatrix}.$$

Let $u = (0, 1, 0)^T \in \mathbb{R}^3$. We then have:

$$H(\theta)u = \lambda_1 \alpha^2 u.$$

The other two eigenvalues are those of the matrix:

$$B = \begin{pmatrix} \lambda_2 \beta_2^2 & 2\lambda_2 \alpha \beta_2 - \lambda_2 \\ 2\lambda_2 \alpha \beta_2 - \lambda_2 & \lambda_2 \alpha^2 \end{pmatrix}.$$

Recall that we have proved in Appendix E.1.1 that when $\alpha \beta_2 < 1$:

$$\lambda_{\max}(B) \le \|B\| \le \lambda_2 \beta_2^2 + \lambda_2 \alpha^2 + \lambda_2.$$

Based on the condition in $\mathbb{D}$ that $\lambda_1 \eta \alpha^2 \ge 1$ and $0 < \alpha \beta_2 < 1$, we have:

$$\frac{\beta_2^2}{\alpha^2} \le \frac{1}{\alpha^4} \le \lambda_1^2 \eta^2.$$

Thus, we obtain:

$$\lambda_{\max}(B) \le \lambda_2 \lambda_1^2 \eta^2 \alpha^2 + \lambda_2 \alpha^2 + \lambda_2$$
$$\le \left( \lambda_1 \lambda_2 \eta^2 + \frac{\lambda_2}{\lambda_1} + \lambda_2 \eta \right) \cdot \lambda_1 \alpha^2$$
$$\le 0.0111 \cdot \lambda_1 \alpha^2.$$

Therefore, $\lambda_1 \alpha^2$ is the largest eigenvalue of $H(\theta)$ and $u$ is its corresponding eigenvector. Thus, $\|H(\theta)\|_2 = \lambda_1 \alpha^2 \le \frac{2}{\eta}$. Recall that the gradient of $\beta_1$ is $2\lambda_1 \alpha^2 \beta_1$, which will be 0 when $\beta_1 = 0$. Therefore, when $\beta_1 = 0$, we have $\nabla L(\theta) \cdot u = 0$. This proves that $\theta$ is in the "stable set".

Now we will prove that the update equation for the constrained trajectory is:

$$\alpha^\dagger(t+1) = \mathbf{Clip}\left( \alpha^\dagger(t) + \eta \lambda_2 \beta_2^\dagger(t)(1 - \alpha^\dagger(t)\beta_2^\dagger(t)), \sqrt{\frac{2}{\eta \lambda_1}} \right), \tag{F.4}$$

$$\beta_2^\dagger(t+1) = \beta_2^\dagger(t) + \eta \lambda_2 \alpha^\dagger(t)(1 - \alpha^\dagger(t)\beta_2^\dagger(t)). \tag{F.5}$$

It is important to note that $\beta_1(t) \equiv 0$ by the definition of $\mathbb{D}$. Based on the definition of $\mathbb{D}$ and the initialization condition, we know $\alpha^\dagger(0)\beta_2^\dagger(0) < 1$. First, we want to prove that if $\alpha^\dagger(t)\beta_2^\dagger(t) < 1$, then we will have (F.4) and (F.5), as well as $\alpha^\dagger(t+1)\beta_2^\dagger(t+1) < 1$. Suppose $\alpha^\dagger(t)\beta_2^\dagger(t) < 1$, then we have:

$$1 - \left( \alpha^\dagger(t) + \eta \lambda_2 \beta_2^\dagger(t) \left( 1 - \alpha^\dagger(t)\beta_2^\dagger(t) \right) \right) \cdot \left( \beta_2^\dagger(t) + \eta \lambda_2 \alpha^\dagger(t) \left( 1 - \alpha^\dagger(t)\beta_2^\dagger(t) \right) \right)$$
$$= \left( 1 - \lambda_2 \eta \left( \alpha^\dagger(t)^2 + \beta_2^\dagger(t)^2 \right) - \lambda_2^2 \eta^2 \alpha^\dagger(t) \beta_2^\dagger(t) \left( 1 - \alpha^\dagger(t)\beta_2^\dagger(t) \right) \right) \left( 1 - \alpha^\dagger(t)\beta_2^\dagger(t) \right).$$

When $\alpha^\dagger(t)\beta_2^\dagger(t) < 1$, based on the definition of $\mathbb{D}$, we have:

$$\lambda_2 \eta \left( \alpha^\dagger(t)^2 + \beta_2^\dagger(t)^2 \right) < \lambda_2 \eta \alpha^\dagger(t)^2 + \frac{\lambda_2 \eta}{\alpha^\dagger(t)^2} \le \frac{2\lambda_2}{\lambda_1} + \lambda_1 \lambda_2 \eta^2 \le 0.011,$$

$$\lambda_2^2 \eta^2 \alpha^\dagger(t) \beta_2^\dagger(t) \left( 1 - \alpha^\dagger(t)\beta_2^\dagger(t) \right) \le \frac{\lambda_2^2 \eta^2}{4} \le 0.001.$$

Thus, we know that if $\alpha^\dagger(t)\beta_2^\dagger(t) < 1$, then

$$\left( \alpha^\dagger(t) + \eta \lambda_2 \beta_2^\dagger(t) \left( 1 - \alpha^\dagger(t)\beta_2^\dagger(t) \right) \right) \cdot \left( \beta_2^\dagger(t) + \eta \lambda_2 \alpha^\dagger(t) \left( 1 - \alpha^\dagger(t)\beta_2^\dagger(t) \right) \right) < 1.$$

It is then apparent that:

$$\mathbf{Clip}\left( \alpha^\dagger(t) + \eta \lambda_2 \beta_2^\dagger(t) \left( 1 - \alpha^\dagger(t)\beta_2^\dagger(t) \right), \sqrt{\frac{2}{\eta \lambda_1}} \right)$$
$$\cdot \left( \beta_2^\dagger(t) + \eta \lambda_2 \alpha^\dagger(t) \left( 1 - \alpha^\dagger(t)\beta_2^\dagger(t) \right) \right) < 1.$$

Therefore, we know that:

$$\left(\mathbf{Clip}\left(\alpha^\dagger(t) + \eta\lambda_2\beta_2^\dagger(t)\left(1 - \alpha^\dagger(t)\beta_2^\dagger(t)\right), \sqrt{\frac{2}{\eta\lambda_1}}\right), 0,\right.$$
$$\left.\beta_2^\dagger(t) + \eta\lambda_2\alpha^\dagger(t)\left(1 - \alpha^\dagger(t)\beta_2^\dagger(t)\right)\right) \in \mathbb{D}.$$

It is then apparent that:

$$\left(\mathbf{Clip}\left(\alpha^\dagger(t) + \eta\lambda_2\beta_2^\dagger(t)\left(1 - \alpha^\dagger(t)\beta_2^\dagger(t)\right), \sqrt{\frac{2}{\eta\lambda_1}}\right), 0,\right.$$
$$\left.\beta_2^\dagger(t) + \eta\lambda_2\alpha^\dagger(t)\left(1 - \alpha^\dagger(t)\beta_2^\dagger(t)\right)\right)$$
$$= \Pi_\mathbb{D}\left(\alpha^\dagger(t) + \eta\lambda_2\beta_2^\dagger(t)\left(1 - \alpha^\dagger(t)\beta_2^\dagger(t)\right), 0,\right.$$
$$\left.\beta_2^\dagger(t) + \eta\lambda_2\alpha^\dagger(t)\left(1 - \alpha^\dagger(t)\beta_2^\dagger(t)\right)\right).$$

Thus, we have shown that if $\alpha^\dagger(t)\beta_2^\dagger(t) < 1$, we have (F.4) and (F.5), as well as $\alpha^\dagger(t+1)\beta_2^\dagger(t+1) < 1$. Therefore, we can prove that for all $t$, (F.4) and (F.5) will hold by induction.

### F.4 Proof of Theorem 5.4

Based on Equation (F.4), Equation (F.5) and $\alpha^\dagger(t)\beta_2^\dagger(t) < 1$, we know that for all $t$ we have:

$$\alpha^\dagger(t+1) \geq \alpha^\dagger(t), \quad \beta_2^\dagger(t+1) > \beta_2^\dagger(t).$$

Since we know that

$$L\left(\theta^\dagger(t)\right) = \frac{1}{2}\lambda_2\left(1 - \alpha^\dagger(t)\beta_2^\dagger(t)\right)^2 > 0,$$

then for all $t$ we have

$$L\left(\theta^\dagger(t+1)\right) < L\left(\theta^\dagger(t)\right).$$

Next, based on the definition of $\widetilde{t}$ and $\alpha^\dagger(t+1) \geq \alpha^\dagger(t)$, we know that for all $t \geq \widetilde{t}$, we have $\alpha^\dagger(t) = \sqrt{\frac{2}{\lambda_1\eta}}$. Therefore, for $t \geq \widetilde{t}$ we have:

$$L\left(\theta^\dagger(t)\right) = \frac{1}{2}\lambda_2\left(1 - \sqrt{\frac{2}{\lambda_1\eta}}\beta_2^\dagger(t)\right)^2,$$
$$\beta_2^\dagger(t+1) = \beta_2^\dagger(t) + \eta\lambda_2\sqrt{\frac{2}{\lambda_1\eta}}\left(1 - \sqrt{\frac{2}{\lambda_1\eta}}\beta_2^\dagger(t)\right).$$

Based on these two equations, it is easy to prove that

$$L\left(\theta^\dagger(t+1)\right) = \frac{1}{2}\lambda_2\left(1 - \sqrt{\frac{2}{\lambda_1\eta}}\beta_2^\dagger(t+1)\right)^2$$
$$= \left(1 - \frac{2\lambda_2}{\lambda_1}\right)^2 \cdot \frac{1}{2}\lambda_2\left(1 - \sqrt{\frac{2}{\lambda_1\eta}}\beta_2^\dagger(t)\right)^2$$
$$= \left(1 - \frac{2\lambda_2}{\lambda_1}\right)^2 \cdot L\left(\theta^\dagger(t)\right).$$

## G Gradient Descent Dynamics for Unconstrained Optimization Problem

We present training results for the original optimization problem without constraints. The experiments use the same initialization set and corresponding learning rate as defined in Section 3. The training dynamics of GD remain consistent with those observed in our constrained optimization setting. For example, in the original problem, with a learning rate of $\eta = \frac{1}{30}$ and an initialization from $\mathcal{X}(\frac{1}{30})$, we trained the model using gradient descent for 15,000 steps. As shown in Figure 11, the results closely resemble the results reported in our main text: the training loss exhibits sharp spikes, and the sharpness displays a recurring pattern of progressive sharpening followed by self-stabilization.

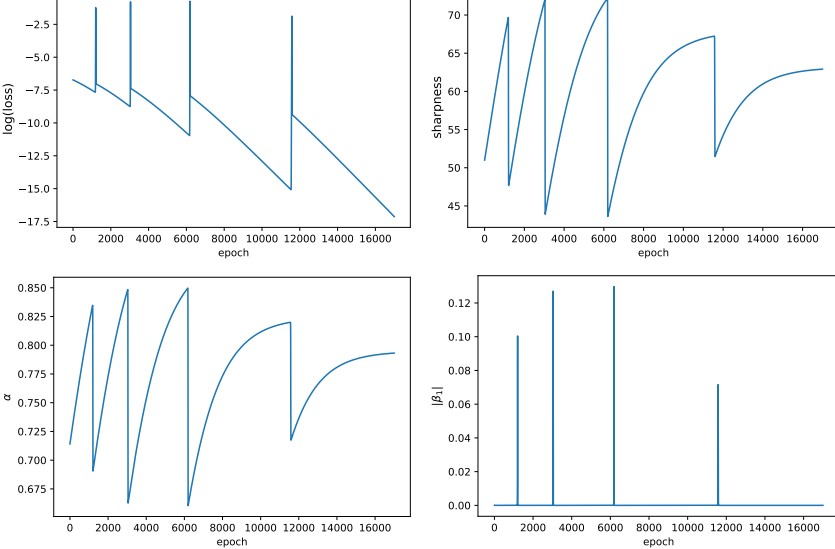

Figure 11: In the original optimization problem, we set $\lambda_1 = 100$ and $\lambda_2 = \frac{1}{100}$. We choose learning rate $\eta = \frac{1}{30}$ and choose a initialization from $\mathcal{X}(\frac{1}{30})$. Then we train the model using GD for 17k steps.

# H  Discussion about the insights we provide

The key difference between our work and Rosenfeld and Risteski [2023] is that Rosenfeld and Risteski [2023] is an empirical study offering insightful but informal hypotheses, while our work provides rigorous theoretical justification for the same phenomena, clarifies previously vague or informal aspects of their explanation, and explains how the behaviors hypothesized in [1] arise.

**Toward Understanding Progressive Sharpening:**  Rosenfeld and Risteski [2023] attempts to link sharpening with "sensitivity" of model, but it remains unclear which quantity precisely drives this sensitivity or why it increases progressively. Our work precisely characterizes the behavior of landscape sharpness in our setting:

- We prove it is tightly controlled by the final-layer $L_2$ norm,
- Sharpness grows linearly with the largest input eigenvalue $\lambda_1$,
- And the $L_2$ norm increases monotonically before a threshold, driving progressive sharpening.

This concretely explains why sharpening occurs and when it halts.

**Why Dynamics Diverge After Initially Converging:**  Rosenfeld and Risteski [2023] speculates that optimizing for the relevant signal eventually amplifies the irrelevant but large scale feature—e.g., the "sky" feature in images, but offers no explanation for why this will happen. We show this phenomenon arises mechanistically in our setting:

- Optimizing the relevant feature increases output-layer parameter $\alpha$,
- Which in turn amplifies the loss on irrelevant dimensions via $\alpha\beta_1$,
- This triggers divergence along the $\beta_1$ direction.

Thus, we offer a quantitative mechanism underlying the divergence pattern observed in Rosenfeld and Risteski [2023].

**Why and How Dynamics Return to Stability:**  Rosenfeld and Risteski [2023] proposes two possible (and incompatible) hypotheses: either the network "downweights the irrelevant feature, flattening the valley" or "catapults to a different basin," without determining which applies and why this will occur. Our analysis proves that:

- Divergence eventually causes $\alpha$ to decrease due to negative gradients induced by large $\beta_1$,

- •This leads to monotonic self-stabilization, which pulls the sharpness back below the threshold,

- • No basin-hopping is needed—our result shows stabilization within the same basin.

Hence, we explain both onset and recovery of instability.

In summary, our work builds a rigorous foundation for understanding Edge-of-Stability through a theoretically grounded minimalist example. We clarify several previously open questions raised in Rosenfeld and Risteski [2023]. Our contribution lies not in proposing entirely new phenomena, but in formalizing and proving mechanisms that were previously only hypothesized. Our framework also explains other empirical observations, such as the monotonic sharpness decrease of GFS in Kreisler et al. [2023], thus offering a unified theoretical lens on EoS dynamics across settings with heterogeneous data distributions.

