# OpenReview forum: "A Minimalist Example of Edge-of-Stability and Progressive Sharpening"
_NeurIPS.cc/2025/Conference — NeurIPS 2025 poster_

### Official Review · Reviewer_EK41 · 2025-06-20

**Clarity:** 2
**Significance:** 3
**Originality:** 3
**Rating:** 5
**Confidence:** 4

**Summary:**

The authors theoretically study optimization of a two-layer, linear, width-1, scalar-output neural network with 2-dimensional inputs using projected gradient descent. The objective is the square population loss, where the data population has covariance with one large eigenvalue corresponding to noise (which plays no role in determining the target value), and one small eigenvalue corresponding to signal (which entirely determines the target value). The authors demonstrate that optimizing this objective with a moderately large learning rate exhibits progressive sharpening, followed by a transient edge of stability phase, followed by monotonic linear convergence of the loss to a global minimum. The authors also compare their setting and results with those of previous work.

**Questions:**

No questions.

**Ethical Concerns:**

["NO or VERY MINOR ethics concerns only"]

**Final Justification:**

I was already convinced that the work was a substantial contribution to the literature on edge of stability. The authors pointed out some ways in which their ideas could be generalised, which I think point to promising directions for future research. My rating remains a 5 (Accept).

**Limitations:**

Limitations (e.g. simplicity of the model) are not explicitly addressed by the authors. If the paper is accepted for publication, I would recommend including a discussion of limitations and directions for future work in the extra space provided.

**Quality:**

3

**Strengths And Weaknesses:**

Strengths:
 - The theory demonstrates all the features of large learning rate training demonstrated in Cohen et al 2021, specifically progressive sharpening and edge of stability. To my knowledge, it is the first to provide properly-grounded guarantees of these phenomena.
 - The theory indicates how the structure of the data plays a role in determining the unstable dynamics.

Weaknesses:
 - The setting is quite limited and simplistic, but no more so than others considered in papers published at venues of equivalent prestige to NeurIPS.
 - The setting studied is projected gradient descent, rather than gradient descent. In this sense, the theory is a departure from the phenomena indicated in Cohen et al.

---

> ### Author Rebuttal · Authors · 2025-07-31
>
> Thank you for your review and for recognizing our work! We believe your evaluation of our paper is very objective. We will do our best to address your concerns below.
>
> **Weakness 1. The setting is quite limited and simplistic, but no more so than others considered in papers published at venues of equivalent prestige to NeurIPS.**
>
> Yes, our setting is a simple example. However, as you mentioned, such minimalist examples are very common at NeurIPS and other top-tier conferences, and our setting is not simpler than those. At the same time, our setting is not a mere replication of previous work, as we introduce a new data distribution that connects to the EoS phenomenon, reproducing EoS behavior closer to practice and providing new insights.
> Moreover, due to the rotational invariance of gradient descent, our setting can be easily extended to high-dimensional inputs with two features.
>
> In addition, there are also some simple methods that allow our setting to be extended to the multi-feature case. We simply change the input from $(x_1, x_2)$ to $(x_1, x_2, x_3)$ where $x_3$ is independent with $x_1, x_2$ and has the same distribution as $x_2$. We also change the target $y$ to be $x_2 + x_3$. Next, we extend the parameters from $(\alpha, \beta_1, \beta_2)$ to $(\alpha, \beta_1, \beta_2, \beta_3)$. Then the loss becomes:
> $$
> \frac{1}{2}\lambda_1(\alpha\beta_{1})^2 +\frac{1}{2}\lambda_2(\alpha\beta_{2}-1)^2 + \frac{1}{2}\lambda_2(\alpha\beta_{3}-1)^2
> $$  Then, by symmetry, if our initialization satisfies $\beta_2(0) = \beta_3(0)$, then for any $t$ we have $\beta_2(t) = \beta_3(t)$, and the loss becomes equivalent to
> $$
> \frac{1}{2}\lambda_1(\alpha\beta_{1})^2 +\frac{1}{2}(2\lambda_2)(\alpha\beta_{2}-1)^2
> $$  which corresponds to the case $\lambda^{'}_1 = \lambda_1$, $\lambda^{'}_2 = 2\lambda_2$ in the original setting with $(\lambda^{'}_1, \lambda^{'}_2)$, and the dynamics are essentially equivalent. Therefore, although our setting is simple, it can in fact be extended to more general cases.
>
> **Weakness 2. The setting studied is projected gradient descent, rather than gradient descent. In this sense, the theory is a departure from the phenomena indicated in Cohen et al.**
>
> We acknowledge that the use of projection introduces a gap between our theoretical setting and practical implementations. However, the optimization dynamics in our setting are discrete and highly unstable, making the theoretical analysis extremely challenging. To make the problem more tractable, we introduce projection as a necessary simplification to enable a rigorous theoretical study. Our numerical experiments show that similar phenomena still occur without projection, indicating that our insights likely generalize beyond the projected setting. Extending the analysis to the unprojected case remains an important but technically demanding direction, which we aim to pursue in future work.
>
> ---
>
> We sincerely thank you again for your recognition of our work! If you have any further questions or concerns, we would be very happy to discuss them with you!

---

> > ### Comment · Reviewer_EK41 · 2025-08-01
> >
> > Thank you for the additional information. Extension of these ideas to more general settings seems a promising direction for future research. My rating remains a 5 (Accept).

---

> > > ### Author Response · Authors · 2025-08-06
> > >
> > > Thank you for your reply and your kind decision! If you have any further questions, we would be more than happy to discuss them with you.

---

### Official Review · Reviewer_pR9a · 2025-07-02

**Clarity:** 4
**Significance:** 2
**Originality:** 3
**Rating:** 4
**Confidence:** 3

**Summary:**

The authors construct a simple, analytically tractable example that exhibits key phenomena observed in deep learning training dynamics, specifically, Edge-of-Stability behavior and Progressive Sharpening when using large learning rates. Their model consists of a two-layer linear network trained with gradient descent on a two-dimensional input, separating task-relevant and irrelevant features. They provide a non-asymptotic, step-by-step analysis demonstrating that the training process self-stabilizes after progressively sharpening: the sharpness of the loss landscape consistently hovers near the critical $2/\eta$ threshold, even as the loss decreases in a non-monotonic fashion. They connect this to notions such as stable set previously established in the literature.

**Questions:**

- Is the study specific to this simple two dimensional example or could it be generalized to larger dimensions and different target functions?
- What happens if the K in line 124 is not very large ? Like for example if we suppose a standard gaussian ?
- Can we conduct such an analysis without supposing a distribution on the data ?
- Is a sharpness of $2/\eta$ considered sharp or flat ?
- The times T1 T2 T3 T4 can be a bit hard to grasp. Can you say for example to which epoch do they correspond on Figure 2?
- In line 377, you say that the learning rate affect the distance between initialization and minima? Is it the case in practice? Aren't the weights initialized independently of the learning rate unlike in this setting ?
- What would change if we use SGD or other optimizers used in practice to train deep neural networks?

Some minor typos:
- line 200 $\alpha$ instead of $\alpha_1$
- line 283, should it be $\phi(\theta(t))$ ?
- line 338 "constrained" instead of "contrained"
- line 368 "spikes" instead of "spiles"

**Ethical Concerns:**

["NO or VERY MINOR ethics concerns only"]

**Final Justification:**

The paper presents good results, but it relies on strong assumptions.

**Limitations:**

yes

**Quality:**

3

**Strengths And Weaknesses:**

Strengths:
- Interesting theoretical study, which explains the EoS phenomena observed in deep learning in a tractable easy-to-interpret setting.
- Clear assumptions and statements and helpful discussions.

Weaknesses:
- The 2D minimalist example, while elegant, may omit complexities in deep, realistic architectures. The main challenge is generalization of these results.
- The study relies on strong assumptions. It is unclear how robust the analysis is to more generic settings.
- The study concerns population loss while in practice we only have access to empirical risk.

---

> ### Author Rebuttal · Authors · 2025-07-31
>
> Thank you for your review and for recognizing our work! We will do our best to answer your questions and address your concerns.
>
> **Question 1. Is the study specific to this simple two dimensional example or could it be generalized to larger dimensions and different target functions?**
>
> Yes, there are many ways to extend our setting to the high-dimensional case. For example, due to the rotational invariance of gradient descent, for any high-dimensional data, as long as there are only two features and these two features satisfy the conditions we mentioned, whether these two features are scalars or high-dimensional vectors, we can use rotational invariance to reduce them equivalently to our setting.
>
> In addition, there are also some methods that allow our setting to be extended to the multi-feature case. For example, we can change the input from $(x_1, x_2)$ to $(x_1, x_2, x_3)$ where $x_3$ is independent with $x_1, x_2$ and have the same distribution with $x_2$. Then we can prove that, in certain cases, the optimization dynamics in this high-dimensional space are fully equivalent to those in our setting.
>
> **Question 2. What happens if the K in line 124 is not very large ? Like for example if we suppose a standard gaussian ?**
>
> In our numerical experiments, we found that similar phenomena generally occur when $K \geq 20$. However, the imbalance in feature scales is the key to the emergence of these phenomena. If the data is changed to standard Gaussian (where $K=1$), the numerical results would be significantly different — in fact, the behavior in that case resembles more close to the results reported in [1].
>
> **Question 3. Can we conduct such an analysis without supposing a distribution on the data ?**
>
> Our analysis can be generalized beyond the Gaussian distribution.
> We only require a distribution with zero mean and covariance matrix
> $\begin{bmatrix}
>     \lambda_1 & 0 \\\\
>     0 & \lambda_2
> \end{bmatrix}$
> to ensure that our loss function remains unchanged.
>
> **Question 4. Is a sharpness of $2/\eta$ considered sharp or flat ?**
>
> In our description, sharp and flat are defined with respect to a fixed learning trajectory $\eta$. We treat $2/ \eta$ as a threshold for $\eta$: sharpness values smaller than $2/\eta$ are considered flat, while those greater than $2/\eta$ are considered sharp. We do not explicitly define the case where the sharpness is exactly equal to $2/\eta$. However, since $2/\eta$ is a single value, the probability that the sharpness exactly equals this value during training is zero. Therefore, the lack of a definition at this single point has no practical impact.
>
> **Question 5. The times T1 T2 T3 T4 can be a bit hard to grasp. Can you say for example to which epoch do they correspond on Figure 2?**
>
> This is a great suggestion — we will include a figure showing $T_1$, $T_2$, $T_3$, and $T_4$ in future versions. Due to the NeurIPS rebuttal constraints, we are currently unable to include the figure. In Figure 1 (we refer to it as Fig. 1 since it is clearer), $T_1$ is approximately at epoch 600, $T_2$ around epoch 2200, $T_3$ around epoch 2215, and $T_4$ around epoch 3100.
>
> **Question 6. In line 377, you say that the learning rate affect the distance between initialization and minima? Is it the case in practice? Aren't the weights initialized independently of the learning rate unlike in this setting ?**
>
> In practice, similar phenomena may indeed occur, and we have some preliminary results supporting this observation. We have conducted some simple experiments by training fully connected networks on CIFAR-10 (with standard initialization methods that are independent of the learning rate). In these experiments, EoS phenomenon will occur, and larger learning rates indeed tend to find solutions that are farther from the initialization. However, verifying whether this holds more broadly may require further practical experiments.
>
> **Question 7. What would change if we use SGD or other optimizers used in practice to train deep neural networks?**
>
> There are some empirical studies on the EOS phenomenon in SGD, which find that small batches stabilize the sharpness at a level much lower than $\frac{2}{\eta}$. Therefore, the EoS of small batches is a very different story. For the EoS phenomenon under other optimizers, even empirical studies are currently limited, and there is a lack of sufficient insight for theoretical analysis. In our current work, we restrict our analysis to full-batch gradient descent, as the optimization dynamics in our setting are already highly unstable and challenging to analyze even in the full-batch regime. However, since our setting is sufficiently simple, it is indeed possible to study the EoS behavior of other optimizers within this framework. We will consider exploring these possibilities in future work.
>
> **Weakness 1. The 2D minimalist example, while elegant, may omit complexities in deep, realistic architectures.**
>
> Please see our answer in **Question 1.**
>
> **Weakness 2. The study relies on strong assumptions. It is unclear how robust the analysis is to more generic settings.**
>
> We acknowledge that our setting is a minimalist example has some assumptions. However, as we demonstrated in our response to Question 1, our setting can be easily extended to more general high-dimensional cases. In addition, we want to argue that some assumptions—such as those on initialization and learning rate—are actually necessary to ensure that the Edge-of-Stability (EoS) phenomenon can occur. Even for practical networks, EoS does not emerge under arbitrary initialization or learning rate choices, and this is even more true in our simplified setting. Moreover, the assumption on the condition number is in fact a core contribution of our work, as we found that the phenomena we demonstrate only emerge under distributions with relatively high condition numbers.
>
> **Weakness 3. The study concerns population loss while in practice we only have access to empirical risk.**
>
> We acknowledge this point; however, we argue that this is a very common trick in theoretical analysis. For example, [1] [2] considers a simpler single-input setting, and [3] also replaces the empirical loss with the population loss in their final theoretical results. Theoretically, when the batch size is sufficiently large, optimization on the empirical loss closely approximates optimization on the population loss.
>
> ---
>
> Thank you again for your review and your recognition of our work! If you have any further questions or concerns, we would be happy to discuss them with you.
>
> **Reference**
>
> [1] Xingyu Zhu and Zixuan Wang and Xiang Wang and Mo Zhou and Rong Ge. Understanding Edge‐of‐Stability Training Dynamics with a Minimalist Example. ICLR 2021.
>
> [2] Itai Kreisler, Mor Shpigel Nacson, Daniel Soudry, and Yair Carmon. Gradient descent monotonically decreases the sharpness of gradient flow solutions in scalar networks and beyond. ICML 2023.
>
> [3] Kwangjun Ahn and Sébastien Bubeck and Sinho Chewi and Yin Tat Lee and Felipe Suarez and Yi Zhang. Learning Threshold Neurons via the “Edge of Stability”. NeurIPS 2023.

---

> > ### Comment · Reviewer_pR9a · 2025-08-07
> >
> > I thank the authors for the answer. I still believe that the assumptions are too strong, and it is not clear how to extend these to more realistic settings beyond some particular cases. The population loss is also a strong assumption, and although theoretically for a very high number of data points the empirical risk approximates the population risk, in practice this number is almost never reached, such that there is usually a discrepancy between the risks. Hence, I will keep my score.

---

> > > ### Author Response · Authors · 2025-08-07
> > >
> > > Thank you for your feedback! Introducing certain assumptions in the analysis is a common limitation of current optimization theory in machine learning. Within this framework, our work aims to provide new insights, and we are actively exploring ways to relax these assumptions and move toward more realistic settings. We thank you again for recognizing our work.

---

### Official Review · Reviewer_mhh3 · 2025-07-03

**Clarity:** 3
**Significance:** 3
**Originality:** 3
**Rating:** 5
**Confidence:** 4

**Summary:**

This paper presents a novel minimalist analysis of progressive sharpening (PS) and edge of stability (EoS) phenomena in gradient descent (GD). Specifically, the authors study a two-layer network of width one with a two-dimensional input consisting of one relevant feature and one dummy dimension. Among existing minimalist examples, this setup most closely reproduces both PS and EoS behaviors, thereby providing a clear insight into the mechanisms behind them. Furthermore, the authors connect their results to existing works (Damian et al., 2022) and (Kreisler et al., 2023), offering a broader perspective on these phenomena.

**Questions:**

- line 72-79: Could you elaborate on this point in more detail? I am having a hard time following the argument. Which part of the main section is this related to?
- line 139: I understand the choice of $c$ being related to the condition number $\lambda_2/\lambda_1$, but why exactly  $c= \frac{1}{2}(\lambda_2/\lambda_1)^{1/4}$?
- line 154: Is choice of the initialization set $\chi(\eta)$ an artifact of the proof? Does this imply that for certain initialization either PS or EoS might not occur?
- line 221: Why is $\tilde{\chi}(\eta)$ more stable than $\chi(\eta)$?
- line 247: I am not following the statement that $L_1$ drops to near zero whenever the GD goes back the stable region.
- line 363: What do you mean by the statement that "large-scale irrelevant feature will provide a regularization"?

**Ethical Concerns:**

["NO or VERY MINOR ethics concerns only"]

**Final Justification:**

While the paper has some limitations, these are well addressed, and the work makes a valuable contribution that merits publication.

**Limitations:**

See weaknesses.

**Paper Formatting Concerns:**

.

**Quality:**

3

**Strengths And Weaknesses:**

**Stregnths**
- Constructing a minimalist example with a large-scale irrelevant feature, inspired by the findings of (Rosenfeld and Risteski, 2023), is both insightful and thought-provoking. The subsequent theoretical analysis confirms that this setting induces both PS and EoS behaviors more completely, unlike the existing minimalist analyses.
- While the result itself may appear limited, the authors strengthen its impact by connecting it to two promising prior analyses of PS and EoS, thereby contributing to a more cohesive understanding of the phenomena. First, they show that GD reduces the sharpness of the gradient flow solution in their minimalist setting, extending the observation made in (Kreisler et al. 2023) for a scalar network. Second, they examine the behavior of GD along the constrained trajectory, inspired by (Damian et al., 2022), where the loss is shown to decrease monotonically. (I would appreciate if the authors could provide a clearer motivation of relating their work to (Kreisler et al. 2023) and (Damian et al., 2022), ideally in the introduction or at the beginning of Section 5.)

**Weaknesses**
- The justification on constraining the $\beta_1$ value appears insufficient, despite some explanations provided by the authors. The statement that "current optimization theories cannot handle instability" while the paper is analyzing instability is somewhat unclear. While the intent seems to be that bounding $\beta_1$ becomes intractable without constraints, this point warrants further elaboration. In fact, it seems that bounding $\beta_1$ could have been a core theoretical contribution.
- The analysis is limited to a specific set of initialization $\chi(\eta)$. It is unclear why this restriction is necessary from a theoretical standpoint. What would happen both practically and theoretically if this constraint was relaxed?

---

> ### Author Rebuttal · Authors · 2025-07-31
>
> Thank you for your review and for recognizing our work! We also appreciate the concerns and questions you have raised. We have carefully read the weaknesses and questions you raised, and we noticed that your question already covers the weaknesses you pointed out. Therefore, we will address your concerns by carefully answering your question.
>
> **Question 1. line 72-79: Could you elaborate on this point in more detail? I am having a hard time following the argument. Which part of the main section is this related to?**
>
> We apologize for not indicating which part of the main text this discussion corresponds to, which may have caused your confusion.
>
> Line 72 -79 mainly corresponds to Sections 5.1 and 5.2 of the main text. In Section 5.1, we prove that on our constructed bivariate input $(x_1, x_2)$, the linear network exhibits the same phenomenon as in [1], namely that the GFS sharpness decreases monotonically.
>
> In Section 5.2, we prove that under our bivariate input setting, the “stable set” assumed and studied in [2] does exist, and the result from [2]—that the loss decreases monotonically and smoothly over this stable set—also holds in our setting. This contrasts sharply with the result in [1], where it was shown that no well-behaved stable set exists under their setting.
>
> **Question 2. line 139: I understand the choice of being related to the condition number $\lambda_2/\lambda_1$, but why exactly $c = \frac{1}{2}(\lambda_2/\lambda_1)^{\frac{1}{4}}$?**
>
> The constant $c$ does not need to be exactly $\frac{1}{2}(\lambda_2/\lambda_1)^{\frac{1}{4}}$, but it should be $O((\lambda_2/\lambda_1)^{\frac{1}{4}})$.
>
> When the dynamic is unstable and $\beta^2_1$ becomes large, we can approximately write the update rule for $\alpha$ as:
>
> $$
> \alpha(t+1) \,\approx\, \alpha(t) - \eta\lambda_1\beta^2_{1}(t)\alpha(t) = (1 - \eta\lambda_1\beta^2_1(t))\alpha(t)
> $$
>
> So when $\beta_1$ reaches the constraint $c$, considering the range of $\eta$, $\alpha$ fast decays at a rate bounded by $1 - c^2\sqrt{\lambda_1/\lambda_2}$. Note that we have already stated in the main text that the sharpness of the Hessian is essentially $\lambda_1\alpha^2$. This brings us to a point that is difficult to handle using existing optimization theory: when the second order Hessian matrix varies quickly, how can we analyze the entire dynamics? We were not able to find a satisfactory method to handle this, so we had to restrict $c^2 \sqrt{\lambda_1 / \lambda_2}$ to be within a constant. This implies that $c^2$ needs to be in $\mathcal{O}(\sqrt{\lambda_2 / \lambda_1})$, and therefore $c$ needs to be in $\mathcal{O}((\lambda_2 / \lambda_1)^{1/4})$.
>
> **Question 3. line 154: Is choice of the initialization set $\mathcal{X}(\eta)$ an artifact of the proof? Does this imply that for certain initialization either PS or EoS might not occur?**
>
> The initialization set $\mathcal{X}(\eta)$ is a technical requirement for our rigorous proofs rather than a fundamental limitation of the phenomenon. Numerical experiments demonstrate that EoS occurs for initializations in a set much larger than $\mathcal{X}(\eta)$. But of course, for certain initializations, either PS or EoS might not occur. For example, if the initialization is directly near a very flat minimal, then GD will simply converge to that minimal; if it is in an extremely sharp region, GD may diverge immediately. We think this should hold for any theoretical or practical model.
>
> **Question 4. line 221: Why is $\tilde{\mathcal{X}}(\eta)$ more stable than $\mathcal{X}(\eta)$?**
>
> Recalling that $\lambda_1\alpha^2$ corresponds to sharpness, in set $\mathcal{X}(\eta)$ we require $\alpha < \sqrt{\frac{2}{\lambda_1\eta}}$, while in set $\tilde{\mathcal{X}}(\eta)$ we further require $\alpha < \sqrt{\frac{1.5}{\lambda_1\eta}}$, which means $\tilde{\mathcal{X}}(\eta)$ is a subset of $\mathcal{X}(\eta)$ with lower sharpness. In optimization theory, regions with lower sharpness are generally considered to be more stable. Therefore, we call $\tilde{\mathcal{X}}(\eta)$ as a more stable subset of $\mathcal{X}(\eta)$.
>
> **Question 5. line 247: I am not following the statement that $L_1$ drops to near zero whenever the GD goes back the stable region.**
>
> This is an empirical observation, though we acknowledge Figure 2 may not clearly illustrate this behavior. We will include a clearer figure in future revisions.
>
> In fact, note that the update equation for $\beta_1$ is given by
>
> $$
> \beta_1(t+1) = (1 - \lambda_1\alpha(t)^2\eta)\beta_1(t)
> $$
>
> where $\lambda_1\alpha(t)^2 \approx S(\theta(t))$, so $\beta_1^2$ decreases rapidly when $S(\theta) < 2/\eta$, which in turn causes $L_1 = \frac{1}{2}\lambda_1\alpha^2\beta^2_1$ to decrease quickly as well.
>
> **Question 6. line 363: What do you mean by the statement that "large-scale irrelevant feature will provide a regularization"?**
>
> The large-scale irrelevant feature $x_1$ and its corresponding parameter $\beta_1$ effectively constrain the sharpness. This is because the update equation for $\alpha$ is
>
> $$
> \alpha(t+1) = \alpha(t) - \eta\lambda_1\beta^2_{1}(t)\alpha(t) + \eta \lambda_2\beta_{2}(t)(1-\alpha(t)\beta_{2}(t))
> $$
>
> so when $\beta_1$ becomes large due to oscillation, it suppresses the growth of $\alpha$, thereby preventing the model from entering regions of high sharpness ($\approx \lambda_1\alpha^2$). In this sense, it plays a regularization-like role. We apologize for not making this point clear. We will add a clarification in this part in the future revision.
>
> **Weakness 1. The justification on constraining the $\beta_1$ value appears insufficient, despite some explanations provided by the authors...**
>
> Please see our answer in **Question 2.**
>
> **Weakness 2. The analysis is limited to a specific set of initialization $\mathcal{X}(\eta)$. It is unclear why this restriction is necessary from a theoretical standpoint. What would happen both practically and theoretically if this constraint was relaxed?**
>
> Please see our answer in **Question 3.**
>
> ---
>
> We sincerely thank you again for your recognition of our work! If you have any further questions or concerns, we would be very happy to discuss them with you!
>
> **Reference**
>
> [1] Itai Kreisler, Mor Shpigel Nacson, Daniel Soudry, and Yair Carmon. *Gradient descent monotonically decreases the sharpness of gradient flow solutions in scalar networks and beyond*. ICML 2023.
>
> [2] Alex Damian, Eshaan Nichani, and Jason D Lee. *Self-stabilization: The implicit bias of gradient descent at the edge of stability*. ICLR 2023.

---

> > ### Comment · Reviewer_mhh3 · 2025-08-05
> >
> > I appreciate the authors' detailed response. I have a follow-up suggestion and a question.
> >
> > As other reviewers have noted, I believe the paper’s message would be clearer if it included a comparison with the case where the relevant part is dominant.
> >
> > Regarding your response to Q2: Is the Hessian in fact reasonably bounded in practice—such that introducing the constraint set $\Theta$ primarily facilitates the analysis while still faithfully reflecting the practical behavior in your setting? I believe this point should be clarified in the paper.

---

> > > ### Author Response · Authors · 2025-08-06
> > >
> > > Thank you for your follow-up! We appreciate your constructive suggestions and will clarify the points you raised accordingly in the next version.
> > >
> > > We agree that including a comparison with the case where the relevant part is dominant is an excellent idea. It would better illustrate why our specific data construction is necessary and highlight the differences between our setting and previous minimalist ones. We will incorporate this comparison in the next version to strengthen the exposition.
> > >
> > > Regarding your question about the constraint set $\Theta$:  Yes, it is primarily introduced to facilitate the theoretical analysis, but still faithfully reflects the practical behavior. We partially addressed this point in lines 148--152 and Appendix F, where we showed that the same phenomenon occurs even without projection (i.e., when gradient descent is run without the constraint set). In our numerical experiments, we have indeed observed that both the Hessian and the rate at which it changes remain reasonably bounded in practice. We will clarify this point further in a future version and support it with empirical results illustrating the range of the Hessian and its variation.
> > >
> > > We sincerely appreciate your helpful suggestion and will revise the paper accordingly to make these points clearer. If you have any further questions or suggestions, we would be very happy to continue the discussion with you!

---

> > > > ### Comment · Reviewer_mhh3 · 2025-08-06
> > > >
> > > > I think we all hope to eventually see a theory that fully explains the phenomenon, but I agree that introducing the projection was necessary in this case—though I cannot be completely certain that it's unavoidable. That said, I believe this paper is strong overall, and I am maintaining my original acceptance score.

---

> > > > > ### Author Response · Authors · 2025-08-07
> > > > >
> > > > > We sincerely thank you for your kind response! We truly appreciate your recognition of the value of our work. Building upon the current results, we are actively exploring avenues toward a more complete theoretical explanation of the phenomenon, and we hope future work will shed further light on this direction.

---

### Official Review · Reviewer_Yj6C · 2025-07-03

**Clarity:** 4
**Significance:** 2
**Originality:** 3
**Rating:** 4
**Confidence:** 4

**Summary:**

This paper presents a simple problem to analytically derive the well-known phenomenon of Edge of Stability (EoS). The authors consider the training of a two-layer linear neural network of width one in a regression task. Under specific settings and assumptions, they show that progressive sharpening happens, followed by EoS, which is divided into two parts: sharpening and self-stabilization, which occur periodically. They also prove the global convergence of gradient descent in the large learning rate regime for this case.

**Questions:**

1) Can you generalize this result for GD without projection? Because in practice, EoS happens even without projection
2) It seems that the problem is symmetric between the relevant and irrelevant parts. Can we switch the dominance such that the relevant part would be much larger?
3) Is it possible to generalize the proofs by making the relevant and irrelevant parts of the input vectors instead of scalars? Here, we will have the same structure of the Hessian, but with blocks. In this way, we can derive a high-dimensional optimization problem that exhibits an EoS.

**Ethical Concerns:**

["NO or VERY MINOR ethics concerns only"]

**Final Justification:**

The paper presents an easy optimization problem that exhibits EoS phenomenon. The authors addressed my questions.

**Limitations:**

Yes

**Quality:**

3

**Strengths And Weaknesses:**

Strengths:
- Well written
- Try to follow the practice
- Good connection with prior work
- A simple problem that exhibits EoS, which one can analyze

Weaknesses:
- Too many constraints on the problem: initialization, condition number, learning rate, projection, etc. Namely, the conditions of the results are very specific, while the EoS happens in a much wider setting. Even in this simple problem.
- Low-dimensional (High-dimensional problems can also be simple)

---

> ### Author Rebuttal · Authors · 2025-07-31
>
> Thank you for your review and for recognizing our work! We will do our best to answer your questions and address your concerns.
>
> **Question 1. Can you generalize this result for GD without projection? Because in practice, EoS happens even without projection?**
>
> The projection is mainly introduced to facilitate our theoretical analysis rather than being fundamental to the phenomenon itself. Given the highly unstable dynamics inherent to EoS, existing theoretical tools are limited, necessitating certain modifications to make rigorous proofs tractable. We are actively exploring more effective theoretical approaches to analyze such instabilities. However, as we discussed in lines 145–152 and Appendix F, the same EoS phenomenon occurs in our setting even without projection.
>
> **Question 2. It seems that the problem is symmetric between the relevant and irrelevant parts. Can we switch the dominance such that the relevant part would be much larger?**
>
> No, if we make such a change, numerically the EoS phenomenon would no longer occur. Due to space limitations, we are unable to provide a complete theoretical proof, but we can briefly outline the intuition as follows: If we switch the dominance such that the relevant part becomes much larger — or more simply, if we swap the scales of $x_1$ and $x_2$, then the loss function becomes:
> $$
> \frac{1}{2}\lambda_2(\alpha\beta_{1})^2 +\frac{1}{2}\lambda_1(\alpha\beta_{2}-1)^2
> $$  where $\lambda_2 \ll \lambda_1$. Now $\beta_1$ becomes the stable direction and $\beta_2$ becomes the sharp direction. By symmetry, we may assume without loss of generality that $\alpha > 0$. Then, when the sharp direction $\beta_2$ starts to oscillate, for every two steps, there is always one step where the gradient of $\alpha$ is negative (the step when $\alpha \beta_2 > 1$). This means that once the dynamic starts to oscillate in the sharp direction, progressive sharpening can no longer occur, which is different from the numerical results in our original setting, where progressive sharpening and self-stabilization occur alternately and continuously.
>
> **Question 3. Is it possible to generalize the proofs by making the relevant and irrelevant parts of the input vectors instead of scalars?**
>
> Yes, it is possible to generalize the proofs by extending the relevant and irrelevant parts of the input vectors from scalars to vectors. Here we provide a simple idea: we change the input from $(x_1, x_2)$ to $(x_1/2, x_1/2, x_2/2, x_2/2)$. Then the Hessian matrix becomes a $4 \times 4$ matrix, which contains two $2 \times 2$ blocks. We can prove that, in certain cases, the optimization dynamics in this high-dimensional space are fully equivalent to those in our setting.
>
> There are also more general ways to extend our setting to high dimension. Due to the rotational invariance of gradient descent, for any high-dimensional data, as long as there are only two features and these two features satisfy the conditions we mentioned, whether these two features are scalars or high-dimensional vectors, we can use rotational invariance to reduce them equivalently to our setting.
>
> **Weakness 1. Too many constraints on the problem: initialization, condition number, learning rate, projection, etc. Namely, the conditions of the results are very specific, while the EoS happens in a much wider setting. Even in this simple problem.**
>
> We fully understand your concern here. The projection in our theory is introduced to facilitate theoretical analysis, as such an optimization discrete and highly unstable process is very difficult to analyze theoretically. As demonstrated in the Appendix F, the dynamics without projection exhibit similar phenomena.
>
> Furthermore, we emphasize that other constraints—such as those on initialization and learning rate—are essential to enable the Edge-of-Stability (EoS) phenomenon. Even in practical neural networks, EoS does not arise under arbitrary initialization or learning rate choices, and this limitation is even more pronounced in our simplified setting.
>
> We want to emphasize that when dealing with such complex and unstable optimization problems, introducing certain constraints to limit the scope of theoretical analysis is often necessary. Moreover, most of our constraints are not strict; they still allow for a considerable range of variability.
>
> **Weakness 2. Low-dimensional (High-dimensional problems can also be simple)**
>
> Please see our answer in **Question 3**.
>
> ---
>
> Thank you again for your review and your recognition of our work! If you have any further questions or concerns, we would be happy to discuss them with you.

---

> ### Comment · Reviewer_Yj6C · 2025-08-08
> **Reply**
>
> Thank you for your responses. I decided to keep my original score (weak accept).
>
> With regards to my third question, your answer somewhat avoids it. It doesn't create additional independent variables, but rather artificially increases the input dimension. However, this is not a crucial issue for my score decision.

---

> > ### Author Response · Authors · 2025-08-08
> >
> > Thank you very much for your kind decision and for recognizing the value of our work!
> >
> > Regarding your third question, we realize we may have initially misunderstood it. Concerning the additional independent variables you mentioned, we are still actively exploring such extensions, but in the meantime, we can provide a simple construction as a supplement to our previous rebuttal:
> >
> > Specifically, we can extend the input from $(x_1, x_2)$ to $(x_1, x_2, x_3)$, where $x_3$ is independent of both $x_1$ and $x_2$ and shares the same distribution as $x_2$. We then redefine the target as $y = x_2 + x_3$. Correspondingly, we extend the parameter vector from $(\alpha, \beta_1, \beta_2)$ to $(\alpha, \beta_1, \beta_2, \beta_3)$. The loss function thus becomes:
> > $$
> > \frac{1}{2}\lambda_1(\alpha\beta_1)^2 + \frac{1}{2}\lambda_2(\alpha\beta_2 - 1)^2 + \frac{1}{2}\lambda_2(\alpha\beta_3 - 1)^2.
> > $$
> >
> > By symmetry, if our initialization satisfies $\beta_2(0) = \beta_3(0)$, it follows that $\beta_2(t) = \beta_3(t)$ for all $t$. Under this condition, the loss simplifies to
> > $$
> > \frac{1}{2}\lambda_1(\alpha\beta_1)^2 + \frac{1}{2}(2\lambda_2)(\alpha\beta_2 - 1)^2,
> > $$
> > which corresponds exactly to the original setting with $\lambda_1' = \lambda_1$ and $\lambda_2' = 2\lambda_2$. The dynamics remain equally equivalent.
> >
> > We understand that this construction may to some extent also be an artifact, so here we present it only as a supplement to our previous rebuttal. However, this construction actually illustrates that adding a new independent variable $x_3$ with a small scale has very similar effects on the dynamics to enlarging the scale of $x_2$. Initializing with $\beta_2(0) \neq \beta_3(0)$ would complicate the analysis, but the essential behavior remains closely aligned with the dynamic in our setting. We will include a discussion of this part in the next version of the paper.
> >
> > Thank you once again for your careful review and for your recognition of our work. If you have any further questions or would like to discuss this in more detail, we would be more than happy to continue the discussion.

---

### Official Review · Reviewer_ApKd · 2025-07-11

**Clarity:** 3
**Significance:** 1
**Originality:** 2
**Rating:** 4
**Confidence:** 4

**Summary:**

The paper analyzes the Edge-of-Stability and Progressive Sharpening phenomena for two-layer networks with a two-dimensional input, a hidden dimension, and an output dimension of width 1.

**Questions:**

# suggestions
1. The paper cites the arXiv version for many papers, and not the conference version. For example, for [1,2,3], the paper cites the arXiv version instead of [1,2,3].

**Ethical Concerns:**

["NO or VERY MINOR ethics concerns only"]

**Final Justification:**

My main concerns were whether the paper introduces a new understanding of the EoS phenomena, and the similarity between this paper and [1]. I have increased my score as the authors have addressed these concerns in their rebuttal.

[1] Elan Rosenfeld and Andrej Risteski. Outliers with opposing signals have an outsized effect on neural network optimization. ICLR 2023.

**Limitations:**

yes

**Quality:**

2

**Strengths And Weaknesses:**

# Strengths
1. The minimalist example the paper analyzes exhibits the Edge-of-Stability phenomenon in a closer way than most other minimalist examples analyzed in other papers.
2. The paper proves that for the minimalist example, many of the phenomena related to the Edge-of-Stability occur. That includes progressive sharpening, self-stabilization, and global convergence.
3. The paper use an approximation for the loss, and shows that the approximation decays at a geometric rate.
4. The paper shows the connection with previous works, namely:
    1. The paper's minimalist example setting is motivated by [1], where they demonstrate that the oscillations during the Edge-of-Stability stage in image classification tasks are driven by “large magnitude” features in the input data. The paper's goal is to demonstrate the Edge-of-Stability in such setting.
    2. They show upper and lower bounds on the GFS sharpness studied in Kreisler et al. [2]. They show that both bounds decrease monotonically, which indicates that the GFS sharpness also decreases monotonically, as demonstrated in Kreisler et al. [2].
    3. They introduce a subset $\mathcal{M}^{\dagger}(\eta)$ of the stable set $\mathcal{M}(\eta)$ studied in Damian et al. [3]. They show that for the projected gradient descent on $\mathcal{M}^{\dagger}(\eta)$, the loss decreases monotonically.

# Weaknesses
1. While the paper does analyze the minimalist example extensively, in my opinion, it does not introduce a new understanding that can be applied to other problems that exhibit the Edge-of-Stability phenomenon.
2. Some of the theorems and lemmas only hold up to some step number. Thus, only partially proving the phenomena related to the Edge-of-Stability.

[1] Elan Rosenfeld and Andrej Risteski. Outliers with opposing signals have an outsized effect on neural network optimization. In International Conference on Learning Representations (ICLR), 2023.
[2] Itai Kreisler, Mor Shpigel Nacson, Daniel Soudry, and Yair Carmon. Gradient descent monotonically decreases the sharpness of gradient flow solutions in scalar networks and beyond. In International Conference on Machine Learning (ICML), 2023.
[3] Alex Damian, Eshaan Nichani, and Jason D Lee. Self-stabilization: The implicit bias of gradient descent at the edge of stability. In International Conference on Learning Representations (ICLR), 2023.

---

> ### Author Rebuttal · Authors · 2025-07-31
>
> Thank you for your review and letting us know your concerns! We will carefully address your concerns and questions below:
>
> **Weakness 1: While the paper does analyze the minimalist example extensively, in my opinion, it does not introduce a new understanding that can be applied to other problems that exhibit the Edge-of-Stability phenomenon.**
>
> We understand your concern, but we believe our paper does introduce broadly applicable insights. As far as we know, our paper is the first theoretical work to establish how data distribution heterogeneity drives the EoS phenomenon.
>
> Specifically, we study the setting where one feature $x_1$ is large-scale but irrelevant, while another $x_2$ is small-scale and relevant. Such construction is essential to reproducing key EOS behaviors including periodic progressive sharpening and self-stabilization, which has **not** been observed in previous minimalist settings (See Section 6.1). More interestingly, we show the decoupled behavior of parameters -- $\beta_1$ associated with $x_1$ oscillates, while $\beta_2$ corresponding to $x_2$ increases monotonically (Lemma 4.2). We also prove how the condition number $\lambda_2 / \lambda_1$ affects the loss decay rate (Theorem 4.6).
>
> These theoretical results offer valuable insights on the central role of data distribution in driving EoS, a factor overlooked in prior minimalist examples where only a scalar input is considered [2,5]. Moreover, our work provides a new perspective for studying EoS in multi-feature settings, and establishes a concrete foundation for extending theoretical understanding beyond current frameworks.
>
> **Weakness 2: Some of the theorems and lemmas only hold up to some step number. Thus, only partially proving the phenomena related to the Edge-of-Stability.**
>
> We acknowledge that some of our results hold for certain steps. This is due to the inherent difficulty of analyzing unstable and discrete optimization dynamics over long horizons. Importantly, this limitation does not compromise our core contributions. We rigorously prove the global convergence of the unstable dynamics and establish an upper bound on the sharpness of the final minimum (Theorem 4.1). We also demonstrate the existence of the EoS phenomenon throughout the entire training process (Theorem 4.3). Additionally, we show that the monotonic decrease of GFS sharpness observed in [2] persists in our example across all steps (Theorem 5.2). Together, these results provide a solid and novel theoretical foundation for understanding EoS.
>
> ---
>
> Thank you again for your review and for your suggestion! If you have any further questions or concerns, we would be happy to address them.
>
> **Reference**
>
> [1] Elan Rosenfeld and Andrej Risteski. Outliers with opposing signals have an outsized effect on neural network optimization. ICLR 2023.
>
> [2] Itai Kreisler, Mor Shpigel Nacson, Daniel Soudry, and Yair Carmon. Gradient descent monotonically decreases the sharpness of gradient flow solutions in scalar networks and beyond. ICML 2023.
>
> [3] Alex Damian, Eshaan Nichani, and Jason D Lee. Self-stabilization: The implicit bias of gradient descent at the edge of stability. ICLR 2023.
>
> [4] Jeremy M. Cohen and Simran Kaur and Yuanzhi Li and J. Zico Kolter and Ameet Talwalkar. Gradient Descent on Neural Networks Typically Occurs at the Edge of Stability. ICLR 2021.
>
> [5] Xingyu Zhu, Zixuan Wang, Xiang Wang, Mo Zhou, Rong Ge. Understanding Edge-of-Stability Training Dynamics with a Minimalist Example. ICLR 2023

---

> > ### Comment · Reviewer_ApKd · 2025-08-05
> >
> > I want to thank the authors for their response.
> >
> > Regarding weakness 1, you have mentioned in the paper that [1] demonstrates that the oscillations during the EoS stage in image classification tasks are driven by “large magnitude” features in the input data. Can you clarify the difference between what [1] demonstrates to what you demonstrate via the minimalist example?
> >
> > [1] Elan Rosenfeld and Andrej Risteski. Outliers with opposing signals have an outsized effect on neural network optimization.

---

> > > ### Author Response · Authors · 2025-08-06
> > >
> > > Thank you for your follow-up. We are happy to clarify the difference between our work and [1], which inspired our study.
> > >
> > > **Key Distinction**
> > > The core difference is that [1] is an empirical study offering insightful but informal hypotheses, while our work provides rigorous theoretical justification for the same phenomena, clarifies previously vague or informal aspects of their explanation, and explains how the behaviors hypothesized in [1] arise.
> > >
> > > ---
> > >
> > > ### 1. Toward Understanding Progressive Sharpening
> > >
> > > [1] attempts to link sharpening with “sensitivity,” loosely defined via spectral norms or output layer growth. However, it remains unclear which quantity precisely drives this sensitivity or why it increases progressively. Their discussion references prior work without pinning down a mechanism.
> > >
> > > In contrast, we precisely characterize sharpness in our setting:
> > >
> > > - We prove it is tightly controlled by the final-layer L2 norm,
> > > - Sharpness grows linearly with the largest input eigenvalue $\lambda_1$,
> > > - And the L2 norm increases monotonically before a threshold, driving progressive sharpening.
> > >
> > > This concretely explains why sharpening occurs and when it halts.
> > >
> > > ---
> > >
> > > ### 2. Why Dynamics Diverge After Initially Converging
> > >
> > > [1] speculates that optimizing for the relevant signal eventually amplifies the irrelevant but large scale feature—e.g., the “sky” feature in images—but offers no explanation for why this will happen.
> > >
> > > We show this phenomenon arises mechanistically in our setting:
> > >
> > > - Optimizing the relevant feature increases output-layer parameter $\alpha$,
> > > - Which in turn amplifies the loss on irrelevant dimensions via $\alpha\beta_1$,
> > > - This triggers divergence along the $\beta_1$ direction.
> > >
> > > Thus, we offer a quantitative mechanism underlying the divergence pattern observed in [1].
> > >
> > > ---
> > >
> > > ### 3. Why and How Dynamics Return to Stability
> > >
> > > [1] proposes two possible (and incompatible) hypotheses: either the network “downweights the irrelevant feature, flattening the valley” or “catapults to a different basin,” without determining which applies and why this will occur. Moreover, they cannot explain why downweighting the irrelevant feature leads to a flatter loss valley.
> > >
> > > Our analysis proves that:
> > >
> > > - Divergence eventually causes $\alpha$ to decrease due to negative gradients induced by large $\beta_1$,
> > > - This leads to monotonic self-stabilization, which pulls the sharpness back below the threshold,
> > > - No basin-hopping is needed—our result shows stabilization within the same basin.
> > >
> > > Hence, we explain both onset and recovery of instability.
> > >
> > > ---
> > >
> > > ### Summary
> > >
> > > In summary, our work builds a rigorous foundation for understanding Edge-of-Stability through a theoretically grounded minimalist example. We clarify several previously open questions raised in [1]. Our contribution lies not in proposing entirely new phenomena, but in formalizing and proving mechanisms that were previously only hypothesized.
> > >
> > > Our framework also explains other empirical observations, such as the monotonic sharpness decrease of GFS in [2], thus offering a unified theoretical lens on EoS dynamics across settings with heterogeneous data distributions.
> > >
> > > We sincerely appreciate your thoughtful engagement. Please let us know if you'd like to continue the discussion.
> > >
> > > ---
> > >
> > > ### Reference
> > >
> > > [1] Elan Rosenfeld and Andrej Risteski. *Outliers with opposing signals have an outsized effect on neural network optimization.* ICLR 2023.
> > > [2] Itai Kreisler, Mor Shpigel Nacson, Daniel Soudry, and Yair Carmon. *Gradient descent monotonically decreases the sharpness of gradient flow solutions in scalar networks and beyond.* ICML 2023.

---

> > > > ### Comment · Reviewer_ApKd · 2025-08-08
> > > >
> > > > I thank the authors for their response.
> > > > I suggest that the authors add a discussion about the difference between your work and [1], either in the appendix or the main part of the paper.
> > > > I have increased my score.
> > > >
> > > > [1] Elan Rosenfeld and Andrej Risteski. Outliers with opposing signals have an outsized effect on neural network optimization. ICLR 2023.

---

> > > > > ### Author Response · Authors · 2025-08-08
> > > > >
> > > > > Thank you very much for your valuable suggestion! We will include a discussion on this in the next version of the paper. We sincerely appreciate your detailed review and the score increase!

---

### Note · Authors · 2025-08-15

**Dear NeurIPS 2025 Reviewers and AC,**

We sincerely thank you for your thoughtful reviews and discussion. We appreciate your recognition of our contributions, including:

- A minimalist, analytically tractable setting reproducing progressive sharpening → edge-of-stability (EoS) → self-stabilization → global convergence, with rigorous non-asymptotic analysis along the **entire GD trajectory**.

- Clear mechanistic links between data heterogeneity and unstable dynamics, with decoupled parameter behavior and a proof that **GFS sharpness decreases monotonically**, connecting and unifying prior analyses of “stable sets” and gradient-flow sharpness.

Below we summarize the main concerns and our responses:

---

**Generality and assumptions** (Reviewers Yj6C, pR9a, EK41)
Our setting is intentionally simple but extensible. Via rotational invariance, the analysis generalizes to higher dimensions with two effective features, and we also provide constructions for multi-feature cases. We will add a section clarifying the scope of our assumptions (initialization, condition number, learning rate) and why they are needed for EoS.

**Projected vs. unprojected GD** (Reviewers Yj6C, mhh3)
We appreciate that reviewers basically agreed that it is reasonable to introduce projection, given the intrinsic difficulty of analyzing the highly unstable dynamic. Projection enables us to make the dynamic analytically tractable. Empirically, the same phenomena occur without projection; we will further emphasize this and clarify the bounding behavior (Hessian and its variation). We hope our work can serve as a starting point to inspire more research on EoS phenomenon from a minimalist perspective.

**Relation to Rosenfeld–Risteski** (Reviewer ApKd)
In the rebuttal we gave a concise subsection contrasting our **formal, mechanism-level** explanation with their empirical hypotheses:
(i) why and when sharpness grows,
(ii) how optimizing the signal amplifies irrelevant large-scale features to induce divergence, and
(iii) why stabilization resumes without basin hopping.
Reviewer ApKd considered our response addressed this concern.

---

We would like to emphasize the paper’s contribution: a clean, provable mechanism for EoS and progressive sharpening that aligns with practice while highlighting the central role of data heterogeneity in unstable large-LR dynamics. Thank you again for your time and consideration.


*Sincerely,*
The Authors

---

### Decision · Program_Chairs · 2025-09-17

**Decision:**

Accept (poster)

**Comment:**

The committee reached a consensus by which this is a strong and insightful submission. The work provides a clean and minimalist study of edge-of-stability and progressive sharpening phenomena, offering both theoretical clarity and empirical demonstrations. Reviewers appreciated the novelty of the construction, the quality of the analysis, and the clarity of the presentation. While some concerns were raised about the scope and broader applicability, the authors effectively addressed many of these in the rebuttal, strengthening confidence in the contribution. Overall, this paper advances our understanding of phenomena of broad interest to the community. I recommend acceptance.